# Approximate Proportionality in Online Fair Division

**Davin Choo** [1] [*] **Winston Fu** [2] [*] **Tzeh Yuan Neoh** [1] [*] **Tze-Yang Poon** [3] [4] [*] **Nicholas Teh** [5] [*]

## Abstract

We study the online fair division problem, where indivisible goods arrive sequentially and must be allocated immediately and irrevocably. Prior work establishes strong impossibility results for approximating classic notions such as envy-freeness up to one good (EF1) and maximin share (MMS) in this setting, but the approximability of proportionality up to one good (PROP1) has remained unresolved. We resolve this gap in two steps. First, we show that three natural greedy allocation rules (standard baselines in fair division) fail to guarantee any multiplicative approximation to PROP1 against an adaptive adversary. These limitations motivate two relaxations: (i) restricting attention to a non-adaptive adversary, and (ii) incorporating coarse predictions in the spirit of learning-augmented algorithms. Under a non-adaptive adversary, we show that the uniform random allocation achieves a meaningful PROP1 approximation with high probability, and this guarantee is essentially tight for this approach; moreover, when item values are sufficiently small, the allocation is near-PROP1 with high probability. Finally, given maximum item value (MIV) predictions, we design an online algorithm that achieves robust approximation guarantees for PROP1, and degrades gracefully under one-sided prediction error. In contrast, we show that EF1, MMS, and PROPX remain inapproximable even with perfect MIV predictions.

## 1. Introduction

Modern machine learning systems routinely make sequential allocation decisions: for instance, routing ad impressions to advertisers, dispatching jobs to workers in cloud platforms, allocating service requests in online marketplaces, and matching drivers to riders. These decisions are inherently *online* (items arrive over time), *irrevocable* (decisions must be committed immediately), and increasingly *prediction-driven* (platforms rely on forecasts of demand, value, or future arrivals). A central challenge is to provide *fairness guarantees* in this setting: the system must commit to allocations without knowing the future, while any predictive signal may be noisy or biased. This tension between irrevocable online decisions, imperfect predictions, and fairness is now a recurring theme across work on online optimization and learning-augmented algorithms (Lykouris & Vassilvitskii, 2021; Mitzenmacher & Vassilvitskii, 2022). Motivated by this perspective, we ask which *individual level* fairness benchmarks remain meaningfully approximable under irrevocability and weak predictive side-information.

We study this problem through the *online fair division* model: indivisible goods arrive sequentially and must be allocated immediately and irrevocably to an agent based on their reported values for the current good. The model has gained visibility in the ML literature through several complementary lenses: (i) prediction-augmented online allocation and scheduling (Purohit et al., 2018; Lattanzi et al., 2020; Zhou et al., 2023; Balkanski et al., 2023; Spaeh & Ene, 2023), (ii) learning-augmented mechanism design and allocation constraints (Agrawal et al., 2022; Gkatzelis et al., 2022; Cohen et al., 2024), and (iii) learning-based approaches to fair division from partial/bandit feedback (Yamada et al., 2024). A key conceptual point in this line of work is that the relevant performance benchmarks are not purely utilitarian: platform designers often seek *individual level* fairness guarantees that remain meaningful even when the instance is adversarial.

In offline fair division, widely used benchmarks include envy-freeness up to one good (EF1) and maximin share fairness (MMS) (Lipton et al., 2004; Budish, 2011; Amanatidis et al., 2017; Kurokawa et al., 2018). In adversarial online settings, however, recent work has established stark impossibilities: for three or more agents, no finite approximation

---

[*]Equal contribution, alphabetical author ordering. We would like to acknowledge Derek Khu who contributed equally to this work and was inadvertently omitted from the author list at submission; see the arXiv version (Choo et al., 2026a). [1]Harvard University, USA [2]Princeton University, New Jersey, USA [3]Centre for Frontier AI Research (CFAR), Agency for Science, Technology and Research (A*STAR), Singapore [4]Institute of High Performance Computing (IHPC), Agency for Science, Technology and Research (A*STAR), Singapore [5]University of Oxford, UK. Correspondence to: Tzeh Yuan Neoh <tzehyuan_neoh@g.harvard.edu>, Nicholas Teh <nicholas.teh@cs.ox.ac.uk>.

*Proceedings of the 43rd International Conference on Machine Learning*, Seoul, South Korea. PMLR 306, 2026. Copyright 2026 by the author(s).

is possible for MMS (Zhou et al., 2023), and similarly no finite approximation is possible for EF1 under comparable informational assumptions (Neoh et al., 2025). These results suggest that, under irrevocability and limited information, many "gold standard" offline notions may be fundamentally unattainable online.

In this work, we ask whether *proportionality*-type guarantees can serve as a meaningful and achievable alternative in the online model. Proportionality requires that each agent obtain at least her proportional fair share of the total value, but with indivisible goods this benchmark can fail even in offline settings. A standard relaxation is *proportionality up to one good* (PROP1), which requires that each agent can reach their proportional share after adding one good outside their own bundle. PROP1 has a long history in offline fair division and related allocation problems (Conitzer et al., 2017; Aziz et al., 2019; 2020; Barman & Krishnamurthy, 2019; Brânzei & Sandomirskiy, 2024). Crucially, PROP1 is implied by EF1 and MMS, so it remains a principled fairness target even when stronger notions are unattainable.

Despite this appeal, the approximability landscape for PROP1 in the online setting has remained unclear. Prior work shows that exact PROP1 may fail to exist online (Benadè et al., 2018; Neoh et al., 2025). Moreover, existing positive results for online PROP1 rely on comparatively strong side-information (e.g., normalization/total-value information) (Neoh et al., 2025). This leaves open a basic question that is central from an ML perspective: under realistic online constraints and weak predictive signals, can we obtain any nontrivial approximation to PROP1?

Motivated by learning-augmented online optimization, we study algorithms that can leverage weak predictions while still targeting provable guarantees. Concretely, we focus on *maximum item value* (MIV) predictions: for each agent, we are given a prediction of the maximum value this agent has for any good. This is substantially weaker than knowing normalization information or the future sequence, yet captures the kind of coarse "scale" information that is routinely produced by forecasting pipelines. We also analyze one-sided (conservative) prediction error, which aligns with the widespread use of upper confidence bounds and safety margins in sequential decision-making (Auer et al., 2002).

### 1.1. Our Contributions and Paper Outline

We initiate a systematic study of approximate PROP1 in online fair division under two settings that are standard in online algorithms and ML: *adaptive adversary* (worst-case feedback/adversarial dependence) and *non-adaptive adversary* (oblivious sequences), and we quantify what is and is not achievable under weak predictions.

**1. Greedy allocation against adaptive adversaries (Section 3).** We show that three natural greedy allocation strategies (representing common heuristics) cannot guarantee any positive approximation to PROP1 against an adaptive adversary. This is notable (and surprising) because greedy rules are pervasive in both theoretical fair division and deployed allocation systems, and because a natural greedy approach can achieve PROP1 under stronger informational assumptions (Neoh et al., 2025). Our results reveal fundamental limits of greedy methods in adversarial online environments: without either randomness against oblivious inputs or predictive information against adaptive adversaries, even the weakest standard proportionality benchmark can be driven to zero.

**2. Random allocation against non-adaptive adversaries (Section 4).** Against a non-adaptive adversary (which commits to the entire input sequence in advance, independent of the algorithm's realized randomness),[1] we prove that the simple algorithm that allocates each good uniformly at random to $n$ agents achieves an $\alpha$-PROP1 guarantee with probability at least $1 - \delta$, where $\alpha = \Theta(1/\log(n/\delta))$ (Theorem 4.1). This is the first known result for the random allocation against non-adaptive adversaries in this setting, requiring neither asymptotic assumptions nor structural constraints. We complement this with a matching lower bound showing this logarithmic dependence is essentially tight for uniform random allocation (Proposition 4.2). Finally, in the common "small items" setting (i.e., when each agent's maximum item value is small relative to their proportional share), the uniformly random allocation achieves near-proportionality: a $(1 - \varepsilon)$-PROP1 guarantee with probability at least $1 - \delta$ (Theorem 4.3). These results provide a useful positive insight: randomness alone, under oblivious inputs, gives us nontrivial and quantifiable fairness guarantees.

**3. Algorithm with MIV predictions against adaptive adversaries (Section 5).** We design an online algorithm that, given MIV predictions, guarantees a $1/n$-PROP1 allocation even against adaptive adversaries (Theorem 5.2). We then show a general *graceful degradation* guarantee under one-sided prediction error: any $\alpha$-PROP1 algorithm for perfect predictions can be transformed to achieve $\alpha(1 - \varepsilon)/(1 - \alpha\varepsilon/n)$-PROP1 under error $\varepsilon$ (Theorem 5.3), giving us an explicit robustness bound for our algorithm (Corollary 5.4). This positions PROP1 as a fairness objective that is compatible with the learning-augmented paradigm: weak predictions materially improve what is achievable in the worst case.

To complete the picture, we prove that three natural and widely studied stronger fairness notions (EF1, MMS, and the stronger proportionality notion PROPX) admit no pos-

---

[1]Such a weakened adversary is commonly studied in online algorithms and enables meaningful probabilistic guarantees.

itive approximation in the online model even with perfect MIV predictions (Proposition 5.1). This strengthens and broadens known impossibilities and underscores that PROP1 is not merely a convenient relaxation: it is a uniquely viable fairness benchmark under weak predictive information.

## 2. Preliminaries and Related Work

We use the notation $\mathbb{N}_+ = \{1, 2, 3, \ldots\}$ to denote the set of positive integers (excluding zero), and $\mathbb{R}_{\geq 0}$ for the set of non-negative real numbers. For any positive integer $z$, we write $[z] := \{1, \ldots, z\}$. For any set $A$, we denote its powerset, the set of all subsets of $A$, by $2^A$.

Our setting involves $n \geq 2$ *agents* and a sequence of $m \geq 1$ indivisible *goods* $G = \{g_1, \ldots, g_m\}$ that arrive online, one at a time. Each good must be allocated immediately and irrevocably to one of the agents upon arrival. Crucially, the total number of goods $m$ is not known in advance and is implicitly revealed only retrospectively once all goods have arrived. We label the goods in their arrival order, and for any time step $t \in [m]$, we define $G^{(t)} := \{g_1, \ldots, g_t\}$ to be the set of the first $t$ goods observed. Thus, $G^{(m)} = G$ denotes the full set once all goods have arrived. Each agent $i \in [n]$ has a non-negative *valuation function* $v_i : 2^G \to \mathbb{R}_{\geq 0}$, which assigns a value to any bundle of goods. Following standard assumptions in fair division, we assume that valuations are *additive*: for any subset $S \subseteq G$, we have $v_i(S) = \sum_{g \in S} v_i(\{g\})$. For convenience, we write $v_i(g)$ as shorthand for $v_i(\{g\})$.

An *allocation* is a tuple $\mathcal{A} = (A_1, \ldots, A_n)$, where each $A_i \subseteq G$ is the bundle allocated to agent $i$. The allocation must form a partition of the goods: the $A_i$ are pairwise disjoint and their union equals $G$.

### 2.1. Fairness Notions

For any time step $t \in [m]$ and agent $i \in [n]$, let $A_i^{(t)}$ denote the bundle held by agent $i$ after the allocation of good $g_t$, with $A_i^{(0)} = \varnothing$ as the initial allocation and $A_i^{(m)} = A_i$ as the final bundle. We are interested in online algorithms that produce allocations satisfying approximate fairness guarantees. In particular, we focus on *proportionality up to one good* (PROP1) and its multiplicative relaxations.

A classical *proportional* allocation ensures that each agent receives utility at least $v_i(G)/n$, but such guarantees are often unachievable even in the offline setting with two agents and one good. We thus focus on the more permissive relaxation of PROP1 in this work, defined as follows.

**Definition 2.1** (PROP1). An allocation $\mathcal{A}$ satisfies *proportionality up to one good* (PROP1) if for each agent $i \in [n]$, either $A_i = G$, or there exists a good $g \in G \setminus A_i$ such that $v_i(A_i \cup \{g\}) \geq \frac{v_i(G)}{n}$.

Unfortunately, it is known that even PROP1 is not always achievable in online settings (Benadè et al., 2018; Neoh et al., 2025). This motivates the study of multiplicative approximations to PROP1 as a more attainable fairness benchmark. Following prior work, we define the notion of $\alpha$-PROP1 for $\alpha \in [0, 1]$: an allocation is $\alpha$-PROP1 if, for each agent $i \in [n]$, either $A_i = G$, or there exists $g \in G \setminus A_i$ such that $v_i(A_i \cup \{g\}) \geq \alpha \cdot \frac{v_i(G)}{n}$. Note that 1-PROP1 is equivalent to PROP1.

We also briefly consider three well-studied, stronger fairness notions: EF1, MMS, and PROPX; note that these are not the focus of this paper, but we define them in order to (later) establish impossibility results and delineate the boundary of what is achievable. These notions are pairwise incomparable but each implies PROP1.

**Definition 2.2** (EF1). An allocation $\mathcal{A} = (A_1, \ldots, A_n)$ is *envy-free up to one good* (EF1) if for every pair of agents $i, j \in [n]$ with $A_j \neq \varnothing$, there exists a good $g \in A_j$ such that $v_i(A_i) \geq v_i(A_j \setminus \{g\})$.

**Definition 2.3** (MMS). Let $\Pi(G)$ denote the set of all $n$-partitions of $G$. The *maximin share* of agent $i \in [n]$ is defined as $\mathsf{MMS}_i := \max_{\mathcal{X} \in \Pi(G)} \min_{j \in [n]} v_i(X_j)$. An allocation $\mathcal{A} = (A_1, \ldots, A_n)$ is then said to be *maximin share* (MMS) fair if $v_i(A_i) \geq \mathsf{MMS}_i$ for all $i \in [n]$.

**Definition 2.4** (PROPX). An allocation $\mathcal{A}$ satisfies *proportionality up to any item* (PROPX) if for each agent $i \in [n]$, we either have $A_i = G$, or $v_i(A_i \cup \{g\}) \geq \frac{v_i(G)}{n}$ for all goods $g \in G \setminus A_i$.

While these stronger fairness notions are attractive, they are known to be inapproximable in online settings under minimal assumptions (Neoh et al., 2025; Zhou et al., 2023).[2] We denote the approximate versions of these fairness notions as $\alpha$-EF1, $\alpha$-MMS, and $\alpha$-PROPX in a similar fashion as $\alpha$-PROP1.

### 2.2. Maximum Item Value (MIV) predictions

Later in Section 5, we study learning-augmented algorithms that have access to useful predictive information about the instance. In particular, we focus on *maximum item value* (MIV) predictions. Given the set of all goods $G$, let $v_i^{\max} = \max_{g \in G} v_i(g)$ denote the maximum value agent $i \in [n]$ assigns to any single good. We assume access to predictions $p_i \approx v_i^{\max}$, and let $\mathbf{p} = (p_1, \ldots, p_n)$ denote the vector of MIV predictions for all agents.

This form of predictive input is considerably weaker than full knowledge of valuations or normalized value vectors, yet it can still meaningfully guide allocation decisions. We

---

[2]PROPX was not explicitly considered in prior work, but the EFX impossibility results from Neoh et al. (2025) extend naturally to PROPX as well.

refer to the case where $p_i = v_i^{\max}$ for all $i \in [n]$ as having *perfect predictions*. This model is in line with prior work that assumes oracle access to predicted information in online fair division (Neoh et al., 2025; Zhou et al., 2023). We also consider settings with one-sided bounded prediction error. Specifically, we assume that the predictions overestimate the true maximum value by at most a factor of $\frac{1}{1-\varepsilon}$ for some $\varepsilon \in [0, 1)$, and never underestimate it.

**Definition 2.5** (MIV Predictions with One-Sided Errors)**.** The prediction vector **p** has one-sided error $\varepsilon \in [0, 1)$ if we have $v_i^{\max} \in [(1 - \varepsilon) \cdot p_i, \ p_i]$ for each agent $i \in [n]$.

This conservative (upper-biased) approach to prediction is motivated by both theory and practice. In online learning and bandit algorithms, for example, upper confidence bounds (UCB) are a common strategy for exploration under uncertainty (Auer et al., 2002). Similarly, in inventory planning and demand forecasting, safety margins are often introduced to guard against shortfalls, favoring overestimation over underestimation. In such settings, underestimating future values or demand is typically more costly than overestimating, making one-sided prediction errors both natural and desirable.

### 2.3. Types of Adversaries in Online Algorithms

In analyzing online algorithms, it is standard to distinguish between two types of adversaries: *adaptive* and *non-adaptive*. An adaptive adversary can observe the algorithm's internal randomness and past decisions, and respond dynamically by selecting future inputs to undermine performance. While adaptive adversaries model worst-case behavior, they often make meaningful fairness guarantees impossible in online allocation settings. In contrast, a non-adaptive adversary commits to the full input sequence in advance, before any of the algorithm's random decisions are made. This restriction allows for meaningful probabilistic guarantees and is widely adopted when studying randomized algorithms.

### 2.4. Other Related Work

We already discussed the two most directly relevant papers, Neoh et al. (2025) and Zhou et al. (2023), which study online fair division with predictive information in essentially the same model as ours. These works assume perfect predictive information. Here we briefly situate our work within broader lines of work that are related to the model more generally.

**Learning-Augmented Online Fair Division.** There is a growing body of work on *learning-augmented algorithms*,[3] which leverage predictive information about future arrivals and analyze the extent to which desirable fairness properties can be achieved, even when predictions may be inaccurate. However, most such works in the context of online fair divi-

---

[3]See Appendix A for additional background on this area.

sion consider a model with *divisible goods*. It is important to note that fairness concepts and the structure of the model differ significantly between divisible and indivisible goods.

In the setting with indivisible goods (which is the focus of our work), Spaeh & Ene (2023) study the allocation of advertising impressions (goods) to advertisers (agents), subject to cardinality constraints on agents' bundle sizes, and with utilitarian social welfare as the objective. Other works such as Balkanski et al. (2023) and Cohen et al. (2024) explore learning-augmented approaches focused on MMS and incentive-compatibility as desirable properties.

**Online Fair Division.** A separate line of work studies online fair division as a purely adversarial online problem. Aleksandrov et al. (2015) study the online fair division problem modeled after a food bank charity problem, where agents are assumed to have binary valuations. They examine fairness properties such as envy-freeness as well as incentive compatibility. Benadè et al. (2018) focus on minimizing envy as an objective, while Zeng & Psomas (2020) build on this by analyzing the trade-off between approximate envy-freeness and a notion of economic efficiency. He et al. (2019) explore a variant of the online model whereby past allocations can be revisited and swapped. Several other works consider an online fair division model with *divisible* goods (Banerjee et al., 2022; 2023; Barman et al., 2022; Huang et al., 2023). For a broader overview of earlier works in online fair division, see the survey by Aleksandrov & Walsh (2020).

## 3. Greedy Fails in General

In this section, we study the natural class of *greedy algorithms*, which aim to allocate each arriving good in a way that attempts to satisfy a given fairness criterion as much as possible at the current timestep. In the classic online setting without predictive information, the greedy approach is both intuitive and arguably the only viable strategy. The key question then is: what properties or (non-predictive) information can be leveraged to guide the greedy algorithm's decisions?

Prior work that incorporates predictive information typically uses it to guide greedy algorithms that achieve approximate fairness guarantees. For instance, Zhou et al. (2023) proposed a greedy algorithm for $n = 2$ that guarantees $1/2$-MMS under normalized valuations. Building on this, Neoh et al. (2025) introduced a different greedy algorithm that satisfies EF1 (and thus also $1/2$-MMS) for $n = 2$, and achieves PROP1 for all $n$.

Given the intuitive appeal and historical success of greedy approaches in related settings, it is natural to consider them as a potential means of achieving PROP1. However, we show that such strategies generally fail to provide any nontrivial approximation to the PROP1 ratio. Specifically, we

analyze three natural greedy allocation strategies and demonstrate that none of them can guarantee $\alpha$-PROP1, for any constant $\alpha > 0$.

To simplify the description of the greedy allocation strategies, we introduce some useful notation. For any timestep $t \in [m]$ and agent $i \in [n]$, let $A_i^{(t)} \subseteq G^{(t)}$ denote the set of goods allocated to agent $i$ after good $g_t$ has been allocated. Let $c_i^{(t)} := \max\{v_i(g) : g \in G^{(t)} \setminus A_i^{(t)}\}$, with $c_i^{(t)} = 0$ if $G^{(t)} \setminus A_i^{(t)} = \varnothing$. Thus, $c_i^{(t)}$ is the value, for agent $i$, of the most valuable arrived good that is not in $i$'s current bundle. We then define the (unnormalized) $\alpha$-PROP1 value for agent $i$ at timestep $t$ as:

$$\alpha_i^{(t)} = \frac{v_i(A_i^{(t)}) + c_i^{(t)}}{v_i(G^{(t)})} \ \text{(or } \infty \text{ if } v_i(G^{(t)}) = 0\text{)}.$$

In other words, the PROP1 ratio at timestep $t$ is simply $\min\{1, n \cdot \min_{i \in [n]} \alpha_i^{(t)}\}$. For the initial state when $t = 0$, we let $G^{(0)} = \varnothing$, $c_i^{(0)} = 0$, and define $\alpha_i^{(-1)} = \infty$. Then, the three greedy allocation strategies considered are as follows.

**Greedy Strategy 1**: At timestep $t$, allocate good $g_t$ to an agent from $\arg\max_{i \in [n]} \frac{v_i(g_t)}{v_i(G^{(t)})}$.

**Greedy Strategy 2**: At timestep $t$, allocate good $g_t$ to an agent from $\arg\min_{i \in [n]} \frac{v_i(A_i^{(t-1)})}{v_i(G^{(t)})}$.

**Greedy Strategy 3**: At timestep $t$, allocate good $g_t$ to an agent from $\arg\min_{i \in [n]} \frac{v_i(A_i^{(t-1)}) + \max\{c_i^{(t-1)}, v_i(g_t)\}}{v_i(G^{(t)})}$.

Intuitively, the first strategy allocates the arriving good $g_t$ to the agent which values it the most, and is commonly used for utilitarian welfare-maximizing guarantees. Meanwhile, the second strategy allocates $g_t$ to the agent which is currently most unsatisfied and is common in offline fair division (e.g., to achieve EFX when valuations are identical in the offline setting (Plaut & Roughgarden, 2018), in the online setting to achieve EF1 for any number of agents when valuations are identical (Neoh et al., 2025; Elkind et al., 2025a); in the fully-informed online fair division setting to get temporal EF1 under generalized binary valuations (Elkind et al., 2025a); and in many other fair division settings). Finally, the third strategy directly attempts to greedily optimize the PROP1 objective itself by allocating $g_t$ to the agent who *would become* the most unsatisfied if *not* given $g_t$.

The following three results show that all of these natural greedy strategies fail to achieve any non-zero approximation ratio to PROP1.

**Proposition 3.1.** *For $n \geq 2$ and any $\alpha > 0$, there exists a sequence of $m$ arriving goods such that the* Greedy Strategy 1 *fails to produce an $\alpha$-PROP1 allocation.*

*Proof.* Suppose $v_i(g_1) = 1$ for all $i \in [n]$. Without loss of generality, suppose $g_1$ is assigned to agent 1. Suppose $v_1(g_t) = 1$ and $v_2(g_t) = \frac{1}{2}$ for all subsequent goods, where $t \in \{2, \ldots, m\}$. For any $t \in \{2, \ldots, m\}$, observe that $\frac{v_1(g_t)}{v_1(G^{(t)})} = \frac{1}{t} > \frac{1/2}{1 + (t-1)/2} = \frac{v_2(g_t)}{v_2(G^{(t)})}$. Thus, agent 2 will *never* receive any good, i.e., $A_2 = \varnothing$. When $m > 1 + 2(\frac{n}{\alpha} - 1)$, we have $\frac{v_2(\varnothing) + v_2(g_1)}{v_2(G)} = \frac{1}{1 + \frac{m-1}{2}} < \frac{\alpha}{n}$. $\square$

**Proposition 3.2.** *For $n \geq 2$ and any $\alpha > 0$, there exists a sequence of $m$ arriving goods such that the* Greedy Strategy 2 *fails to produce an $\alpha$-PROP1 allocation.*

*Proof.* Suppose $v_i(g_1) = 1$ for all $i \in [n]$. Choose $m \geq \lceil 2n/\alpha \rceil + 1$. Without loss of generality, suppose $g_1$ is assigned to agent 1. Suppose $v_1(g_t) = 1$ and $v_2(g_t) = \frac{1}{m^2}$ for all subsequent goods, where $t \in \{2, \ldots, m\}$. For any $t \in \{2, \ldots, m\}$, observe that $\frac{v_2(A_2)}{v_2(G^{(t)})} \leq \frac{\frac{t-1}{m^2}}{1 + \frac{t-1}{m^2}} = \frac{t-1}{m^2 + t - 1} < \frac{1}{t} = \frac{v_1(A_1)}{v_1(G^{(t)})}$. Thus, agent 1 will *never* receive any subsequent good, i.e., $A_1 = \{g_1\}$. When $m > \frac{2n}{\alpha}$, we have $\frac{v_1(g_1) + v_1(g_2)}{v_1(G)} = \frac{2}{m} < \frac{\alpha}{n}$. $\square$

**Proposition 3.3.** *For $n \geq 2$ and any $\alpha > 0$, there exists a sequence of $m$ arriving goods such that* Greedy Strategy 3 *fails to produce an $\alpha$-PROP1 allocation.*

*Proof sketch.* The construction is rather complicated; we outline the key ideas for the case $n = 2$ and defer the full details to the appendix. In our construction, we maintain that $\alpha_1^{(t)} \neq \alpha_2^{(t)}$ for all $t \geq 3$. Then, fix any timestep $t \geq 4$ and suppose without loss of generality that $\alpha_1^{(t-1)} < \alpha_2^{(t-1)}$. Let $\zeta := c_2^{(t-1)}/(2v_2(G^{(t-1)}))$. We consider two cases.

**Case 1:** $\alpha_2^{(t-1)} > \alpha_1^{(t-1)}(1 + \zeta)$. Then we construct a sequence of $\tau \geq 1$ arriving goods, each worth $c_2^{(t-1)}/2$ to agent 2 and 0 to agent 1 (and small enough so that $c_1, c_2$ do not change). We then show that Greedy Strategy 3 must allocate all these goods to agent 1. Thus, $\alpha_1^{(t-1+\tau)} = \alpha_1^{(t-1)}$, while $v_2(G)$ increases and so $\alpha_2$ decreases, until $\alpha_1^{(t-1)} < \alpha_2^{(t-1+\tau)} \leq \alpha_1^{(t-1)}\left(1 + \frac{c_2^{(t-1)}}{2v_2(G^{(t-1+\tau)})}\right) \leq \alpha_1^{(t-1)}(1 + \zeta)$. That is, the PROP1 ratio (the minimum $\alpha$) stays the same, but the gap shrinks to within a $(1 + \zeta)$ factor.

**Case 2:** $\alpha_2^{(t-1)} \leq \alpha_1^{(t-1)}(1 + \zeta)$. Define the arriving good $g_t$ by $v_1(g_t) = c_1^{(t-1)}$ and $v_2(g_t) = c_2^{(t-1)}$. Under Greedy Strategy 3, regardless of whether $g_t$ is allocated to agent 1 or 2, we have $\min\{\alpha_1^{(t)}, \alpha_2^{(t)}\} < \alpha_1^{(t-1)}$: if $g_t$ is given to agent 2 then $\alpha_1$ strictly decreases since its numerator stays fixed while $v_1(G)$ increases; if $g_t$ is given to agent 1 then $\alpha_2$ strictly decreases by a factor $v_2(G^{(t-1)})/(v_2(G^{(t-1)}) + c_2^{(t-1)})$, and the assumption $\alpha_2^{(t-1)} \leq \alpha_1^{(t-1)}(1+\zeta)$ implies the new $\alpha_2$ drops below $\alpha_1^{(t-1)}$.

By alternating between these two cases, we obtain an infinite sequence along which the minimum $\alpha^{(t)}$ decreases without bound, so for any target $\alpha > 0$ some finite prefix yields an allocation that is not $\alpha$-PROP1. The extension to $n > 2$ keeps agents $3, \ldots, n$ inactive (value $0$ for the constructed goods), so they never affect the greedy choice. □

These negative results highlight the fundamental limitations of greedy allocation strategies when facing adaptive adversaries. Thus, in the next two sections, we explore two approaches to overcome these barriers. In Section 4, we show that a random allocation strategy can achieve a non-trivial PROP1 approximation against non-adaptive adversaries. In Section 5, we show that access to MIV predictions enables non-trivial PROP1 approximations even against adaptive adversaries.

## 4. Random Allocations

We investigate the approximate PROP1 guarantee achieved by the simple algorithm RAND, which allocates each arriving good to an agent uniformly at random. Lipton et al. (2004) analyzed this algorithm in the offline fair division setting, for its incentive-compatible property (i.e., no agent will have the incentive to misreport their valuations so as to obtain a strictly better outcome). Moreover, the uniformly random allocation rule is also frequently used as a benchmark in the fair division literature, either as a point of comparison for fairness guarantees or as a baseline for empirical evaluation (Babichenko et al., 2024; Basteck, 2018; Hosseinzadeh Ranjbar & Feizi, 2023; Nesterov, 2017; Yamada et al., 2024). Given the limited understanding of how to guarantee PROP1 in the online setting, this baseline provides especially valuable insights.

As mentioned earlier in the paper, to the best of our knowledge, this is the first multiplicative approximation bound for PROP1 in the online setting that holds *without* any additional assumptions (such as asymptotic behavior, specific valuation classes, or constraints on the arrival order of items). Achieving such a guarantee is inherently challenging under adaptive adversaries (who tailor their inputs based on the algorithm's random decisions), often forcing worst-case outcomes that render fairness guarantees vacuous. In contrast, studying *non-adaptive* adversaries offers a realistic and tractable framework for evaluating randomized allocation rules in adversarial environments.

We analyze the performance of RAND against a non-adaptive adversary. By linearity of expectation, for each agent $i \in [n]$, we have $\mathbb{E}[v_i(A_i)] = v_i(G)/n$, i.e., RAND is proportional. However, this does not give us meaningful guarantees for the realized allocation. A more meaningful perspective is to study the tail guarantee of RAND: fixing a failure probability $\delta > 0$, we ask: what is the largest $\alpha$

such that RAND guarantees an $\alpha$-PROP1 allocation with probability at least $1 - \delta$?

**Theorem 4.1.** *Fix any $\delta \in (0, 1)$ and $n \geq 2$ agents. Against a non-adaptive adversary,* RAND *achieves $\alpha$-PROP1 with probability at least $1 - \delta$, where $\alpha = \Theta\left(1/\log(n/\delta)\right)$.*

*Proof idea.* Fix any agent $i \in [n]$ and let $\alpha = \frac{3}{32 \log(n/\delta)}$. If some single good $g$ is already worth at least $\alpha \cdot \frac{v_i(G)}{n}$ to agent $i$, then this is sufficient to guarantee that $v_i(A_i \cup \{g\}) \geq \alpha \cdot \frac{v_i(G)}{n}$. Thus, the only interesting case is when every good is "small", i.e., $v_i(g_j) < \alpha \cdot \frac{v_i(G)}{n}$ for all $j \in [m]$. Since under RAND, every good is allocated to $i$ with probability $\frac{1}{n}$, let $S_i'$ be the total value that other agents receive. Since the goods are small, $S_i'$ is a sum of independent, bounded terms whose mean is $\frac{n-1}{n} \cdot v_i(G)$ with variance $\frac{\alpha \cdot v_i(G)^2}{n^2}$.

A careful application of Bernstein's inequality then implies that the probability that $S_i'$ exceeds its mean by $\frac{1-\alpha}{n} \cdot v_i(G)$ is at most $\delta/n$. Applying a union bound over all $n$ agents, the probability that *any* agent fails this guarantee is at most $\delta$. Thus, with probability $1 - \delta$, the final allocation is $\alpha$-PROP1. □

We complement the above result with a matching upper bound which shows that this logarithmic dependence is essentially unavoidable for RAND.[4]

**Proposition 4.2.** *Fix any $\delta \in (0, 1/2]$ and $n \geq 2$ agents. Against a non-adaptive adversary, there exists an instance on which* RAND *returns an allocation that is not $\alpha$-PROP1 with probability at least $\delta$, for some $\alpha = \Theta(1/\log(n/\delta))$.*

The above result shows that the logarithmic dependence on $\log(n/\delta)$ in Theorem 4.1 is essentially tight for RAND.[5] Intuitively, the lower bound exploits the possibility that, with non-negligible probability, some agent receives too little total value for any single additional good to compensate. On the other hand, when every individual good is sufficiently small relative to an agent's proportional share, RAND exhibits much stronger concentration around its expectation. The following theorem formalizes this "small items" setting and shows that RAND then achieves an almost proportional outcome with high probability.

**Theorem 4.3.** *Fix any $\delta \in (0, 1)$, $n \geq 2$, and $\varepsilon \in (0, 1)$. Suppose that for every agent $i \in [n]$, $\max_{g \in G} v_i(g) \leq$*

---

[4]Note that only for the following result, we will use $\alpha$-PROP1 for $\alpha > 1$. This would only correspond to a stronger benchmark than PROP1.

[5]This does not preclude other online algorithms from achieving stronger high-probability guarantees against a non-adaptive adversary; indeed, even in the online setting, Elkind et al. (2025a, Thm 3.7) showed that exact EF1 (and hence PROP1) is achievable under identical valuations. Characterizing the optimal high-probability approximation achievable by general online algorithms against non-adaptive adversaries remains an interesting direction.

$\frac{3\varepsilon^2}{8\log(n/\delta)} \cdot \frac{v_i(G)}{n}$. *Then, against a non-adaptive adversary,* RAND *achieves* $(1 - \varepsilon)$-PROP1 *with probability at least* $1 - \delta$.

## 5. Maximum Item Value (MIV) Predictions

In this section, we study how maximum item value (MIV) predictions can help achieve approximate PROP1 allocations online. Prior work obtains online PROP1 under stronger informational assumptions, such as normalization information about agents' total values (Neoh et al., 2025). Such information can be difficult to obtain in online settings, since it requires estimating an agent's total value over the whole future sequence. In contrast, MIV predictions require *significantly less information*. For each agent, we assume access only to a (possibly approximate) prediction of the value of their most-preferred good among the entire sequence of arriving goods. This represents a more "lightweight" and realistic informational assumption in the online setting.

We note that MIV predictions are less demanding than normalization information. The results of Neoh et al. (2025) imply that while PROP1 is always achievable under normalization information, it is not always achievable under MIV predictions; the latter is a conclusion that can be derived from their analysis following Theorem 3.3. Similarly, while EF1 and $1/2$-MMS are achievable for $n = 2$ given normalization information (Neoh et al., 2025; Zhou et al., 2023), the following proposition shows that neither EF1 nor MMS can be approximated (even for $n = 2$) when only MIV predictions are available. For completeness, we also include results for PROPX, a natural strengthening of PROP1 and another relaxation of proportionality (see Definition 2.4).

**Proposition 5.1.** *For $n \geq 2$ and any $\alpha > 0$, no online algorithm can always return an allocation that is $\alpha$-EF1, $\alpha$-MMS, or $\alpha$-PROPX, even with perfect MIV predictions.*

Given these relatively lightweight predictions, we first show that under accurate MIV predictions, it is possible to design an online algorithm that achieves a finite approximation ratio for PROP1, which scales inversely with the number of agents. Without loss of generality[6], we assume $\mathbf{p} = (p_1, \dots, p_n) = (1, \dots, 1)$; for arbitrary $\mathbf{p}$, one can normalize the valuations by dividing each $v_i(g_t)$ by $p_i$, which preserves the structure of the problem. We introduce the following algorithm (Algorithm 1) that is based on the following potential function, for each agent $i \in [n]$:

$$\phi_i^t = \frac{a_i^{(t)}}{(n^2 + n + 1) \cdot a_i^{(t)} + n^2 \cdot v_i(A_i \setminus \{g_{r_i}\}) \cdot a_i^{(t)} - 1}.$$

[6]This is because $\alpha$-PROP1 is homogeneous in each agent's valuation scale. Dividing all of agent $i$'s valuations by $p_i$ scales both sides of the $\alpha$-PROP1 inequality by the same factor and simultaneously normalizes the predicted maximum to 1, so the guarantee, feasibility and approximation ratio are unchanged.

Before presenting the guarantee, we explain the role of the variables in the potential function. The variable $r_i$ is the first time at which agent $i$ sees a good whose normalized value is 1; before this happens, $r_i = \infty$. The good $g_{r_i}$, once it arrives, accounts for the "one good" allowed in the PROP1 comparison: if it is outside $A_i$, it can be added to $i$'s bundle, and if it is already in $A_i$, then $i$'s bundle already contains that value.

The term $a_i^{(t)}$ scales the total value that must be covered after allowing for this one good. Before $r_i$ occurs, the algorithm reserves room for a future value-1 good by using $a_i^{(t)} = 1/(1 + v_i(G^{(t)}))$. After $r_i$ occurs, that good is known, and the potential uses $A_i^{(t)} \setminus \{g_{r_i}\}$. A smaller potential means that agent $i$ is farther from violating the $1/n$-PROP1 target. Next, the potential function $\phi_i^t$ measures how close agent $i$ is to violating the $\frac{1}{n}$-PROP1 guarantee. A lower potential value indicates a more favorable state. We visualize how $\phi_i^t$ behaves with respect to different values of $a_i^{(t)}$ and the product $v_i(A_i \setminus \{r_i\}) \cdot a_i^{(t)}$.

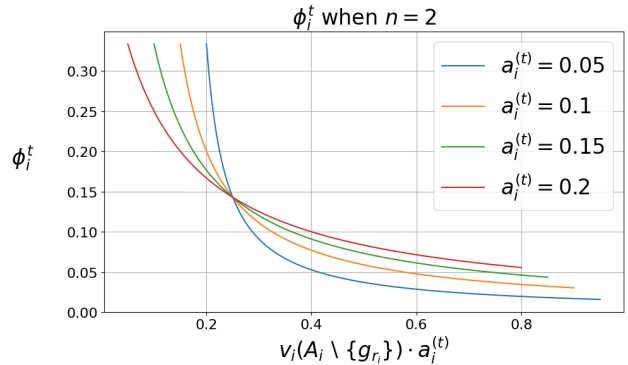

*Figure 1.* $\phi_i^t$ at different values of $a_i^{(t)}$ and $v_i(A_i \setminus \{r_i\}) \cdot a_i^{(t)}$.

This potential function captures two key intuitions: (1) when $v_i(A_i \setminus \{r_i\}) \cdot a_i^{(t)}$ is large, it is better for $a_i^{(t)}$ to be small, indicating that future goods are relatively small compared to $v_i(G)$, and thus will not significantly reduce agent $i$'s surplus; (2) conversely, when $v_i(A_i \setminus \{r_i\}) \cdot a_i^{(t)}$ is small, it is preferable for $a_i^{(t)}$ to be large, ensuring that the algorithm does not need to allocate a low-valued good to agent $i$ merely to avoid violating the $\frac{1}{n}$-PROP1 condition.

Then, our result is as follows.

**Theorem 5.2.** *Given the MIV predictions $\mathbf{p}$, Algorithm 1 returns a $\frac{1}{n}$-PROP1 allocation.*

*Proof sketch.* Let the global potential function be $\Phi^t = \sum_{i \in [n]} \phi_i^t$. The proof relies on two key properties of the potential function, both of which directly lead to the result:

1. If $\Phi^m \leq \Phi^0 = \frac{1}{n+1}$, then the allocation $\mathcal{A}$ returned by

---

**Algorithm 1** Returns a $\frac{1}{n}$-PROP1 allocation given MIV predictions $\mathbf{p} = (p_1, \ldots, p_n) = (1, \ldots, 1)$

---

1: Initialize the empty allocation $\mathcal{A} = (A_1, \ldots, A_n)$ where $A_i \leftarrow \varnothing$ for each $i \in [n]$
2: $r_i \leftarrow \infty$ for each $i \in [n]$
3: **while** there exists a good $g_t$ arriving online **do**
4:     **for** each agent $i \in [n]$ **do**
5:         **if** $v_i(g_t) = 1$ and $r_i > t$ **then**
6:             $r_i \leftarrow t$
7:         **end if**
8:         **if** $t < r_i$ **then**
9:             $a_i^{(t)} \leftarrow 1/(1 + v_i(G^{(t)}))$,
10:            $b_i^{(t)} \leftarrow a_i^{(t)}/((n^2 + n + 1) \cdot a_i^{(t)} + n^2 \cdot v_i(A_i) \cdot a_i^{(t)} - 1)$
11:            $c_i^{(t)} \leftarrow a_i^{(t)}/((n^2 + n + 1) \cdot a_i^{(t)} + n^2 \cdot v_i(A_i \cup \{g_t\}) \cdot a_i^{(t)} - 1)$
12:        **else**
13:            $a_i^{(t)} \leftarrow 1/v_i(G^{(t)})$,
14:            $b_i^{(t)} \leftarrow a_i^{(t)}/((n^2 + n + 1) \cdot a_i^{(t)} + n^2 \cdot v_i(A_i \setminus \{g_{r_i}\}) \cdot a_i^{(t)} - 1)$
15:            $c_i^{(t)} \leftarrow a_i^{(t)}/((n^2 + n + 1) \cdot a_i^{(t)} + n^2 \cdot v_i(A_i \cup \{g_t\} \setminus \{g_{r_i}\}) \cdot a_i^{(t)} - 1)$
16:        **end if**
17:    **end for**
18:    Let $i^* \in \arg\min_{i \in [n]} \left( c_i^{(t)} + \sum_{j \in [n], j \neq i} b_j^{(t)} \right)$
19:    $A_{i^*} \leftarrow A_{i^*} \cup \{g_t\}$
20: **end while**
21: **return** $\mathcal{A} = (A_1, \ldots, A_n)$

---

Algorithm 1 satisfies $\frac{1}{n}$-PROP1.

2. For every $t \in [m]$, there always exists an agent to whom $g_t$ can be allocated such that $\Phi^t \leq \Phi^{t-1}$.

Since the algorithm selects, at each step, the agent to allocate $g_t$ to, such that $\Phi^t$ is minimized, the second property guarantees that the potential never increases: $\Phi^t \leq \Phi^{t-1}$ for all $t$, and thus $\Phi^t \leq \Phi^0$. Combined with the first property, this implies the desired $\frac{1}{n}$-PROP1 guarantee.

To show the second property, let $\phi_i^+$ denote the increase in agent $i$'s potential function if they do *not* receive $g_t$, and let $\phi_i^-$ be the decrease in their potential if they do receive $g_t$. The key idea is to show that for every agent $i$, $(n-1) \cdot \phi_i^+ \leq \phi_i^-$. This inequality implies that the decrease in potential for at least one agent (upon receiving $g_t$) outweighs the total increase in potential across all other agents if they were not allocated the good. Hence, at each step, there always exists an agent whose allocation of $g_t$ ensures the global potential does not increase. $\square$

To the best of our knowledge, the above result gives the first finite multiplicative approximation guarantee known for any proportionality-type benchmark in this adversarial online setting under such weak predictive information. Determining the optimal dependence on $n$, even with access to these predictions, remains an interesting open problem.

Next, we extend our results to account for one-sided error[7] in the MIV predictions, i.e., when each agent's predicted MIV overestimates their true maximum value. The following theorem shows that any algorithm designed for perfect predictions can be made robust to such errors, while degrading gracefully in its guarantees. Note that the approximation ratio achieved is non-trivial as long as $\varepsilon < 1$ and smoothly recovers the original approximation ratio when $\varepsilon = 0$.

**Theorem 5.3.** *Let $\alpha \in [0, 1]$ and suppose there exists an online algorithm that, given the perfect MIV predictions, always outputs an $\alpha$-PROP1 allocation. For any $\varepsilon \in [0, 1)$, there is an online algorithm which, given the MIV predictions whose one–sided error is at most $\varepsilon$, always returns a $\beta$-PROP1 allocation with $\beta = \alpha(1 - \varepsilon)/(1 - \frac{\alpha\varepsilon}{n})$.*

By combining Theorem 5.2 and Theorem 5.3, we obtain the following corollary, which provides a robustness guarantee for our algorithm under one-sided prediction error.

**Corollary 5.4.** *For any $\varepsilon \in [0, 1)$, there is an online algorithm which, given the MIV predictions whose one–sided error is at most $\varepsilon$, always returns a $\beta$-PROP1 allocation with $\beta = \frac{(1-\varepsilon)}{(n-\frac{\varepsilon}{n})}$.*

---

[7]Our one-sided results can be viewed as the calibrated form of bounded two-sided noise via the following reduction. Suppose the raw predictor satisfies $p_i \in [(1 - \rho)v_i^{\max}, (1 + \rho)v_i^{\max}]$ for each agent $i$. Conservatively inflating the prediction to $\tilde{p}_i = p_i/(1 - \rho)$ yields a one-sided error with $\varepsilon = 2\rho/(1 + \rho)$.

**Discussion.** Standard lower bound constructions in online fair division rely on hiding whether a large future item exists (Zhou et al., 2023; Neoh et al., 2025). However, MIV predictions reveal precisely the scale of any future item value, which breaks standard hardness constructions for PROP1. Any tight lower bound would therefore need to fix the entire MIV vector while encoding hardness in the residual tail and arrival order, which is substantially more subtle. Thus, Theorem 5.1 provides a strong partial lower bound that even with perfect MIV predictions, stronger notions (EF1, MMS, PROPX) remain inapproximable, suggesting that our positive result is highly specific to PROP1.

**Empirical comparisons.** In addition to our theoretical analysis, we empirically compared Algorithm 1 with the three greedy baselines described in Section 3.[8]

We first normalize each agent's values by her maximum item value, as in the MIV-normalized view of Algorithm 1. For an allocation $A$, we measure fairness by the realized PROP1 ratio $\min_{i \in [n]} \min\{1, n(v_i(A_i) + \max_{g \in G \setminus A_i} v_i(g))/v_i(G)\}$, with ratio 1 for agents with $v_i(G) = 0$ or $A_i = G$, and measure efficiency by utilitarian welfare normalized by the ex-post optimum, $\sum_i v_i(A_i) / \sum_{g \in G} \max_i v_i(g)$. We use $n = 8$, $m = 40$, and 500 independent trials for each of four instance families: iid uniform values; dense binary-interest instances, where each agent values each good with probability 0.6 and positive values lie in $[0.8, 1]$; correlated values of the form $0.5 U_g + 0.5 U_{i,g}$; and specialist instances, where each good has one high-value specialist and all other agents have small residual values. Error bars in Figure 2 are 95% confidence intervals.

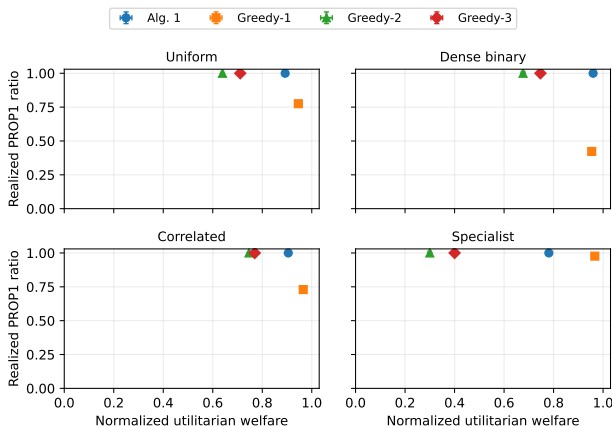

*Figure 2.* Fairness-efficiency tradeoff for Algorithm 1 and the greedy baselines. Algorithm 1 achieves PROP1 on all sampled random instances and dominates Greedy-2 and Greedy-3 in welfare on these distributions. Greedy-1 often has high welfare, but its PROP1 ratio can be substantially lower on competitive instances.

The results are consistent with our theory. Algorithm 1 achieved PROP1 on every sampled instance. Its normalized welfare was 0.892, 0.960, 0.905, and 0.781 on the uniform, dense, correlated, and specialist families, respectively. Greedy-1 achieved welfare 0.946, 0.954, 0.965, and 0.967 on the same families, but with realized PROP1 ratios 0.775, 0.423, 0.730, and 0.977. Thus, Greedy-1 often attains high welfare, but its PROP1 ratio can fall substantially; in the uniform and correlated families this reflects a welfare–fairness tradeoff, while in the dense family Algorithm 1 slightly improves both welfare and fairness. On easy specialist instances, Greedy-1 has higher welfare with little fairness loss. Greedy-2 and Greedy-3 were at or near PROP1 on these benign random instances, but their welfare was substantially lower: 0.639/0.711 on uniform instances, 0.677/0.747 on dense instances, 0.746/0.770 on correlated instances, and 0.300/0.400 on specialist instances. We also stress-tested the explicit counterexample families from Section 3. On prefixes of length roughly 500, the targeted greedy baselines have realized PROP1 ratios 0.008, 0.008, and 0.385 for the Greedy-1, Greedy-2, and Greedy-3 constructions, respectively, while Algorithm 1 obtains 0.920, 1.000, and 1.000 on the same prefixes.

## 6. Conclusion

In this work, we initiate a study of approximate proportionality in online fair division, a setting where many classic fairness notions admit strong impossibility results. Our results present a clear boundary between impossibility under adversarial uncertainty, and what becomes achievable with modest relaxations. On the negative side, we show that several natural greedy baseline algorithms can fail to guarantee any multiplicative approximation to PROP1 against an adaptive adversary, highlighting the challenges of online decision making under worst-case arrivals. On the positive side, we identify two complementary paths to robust guarantees: (i) under a non-adaptive adversary, we show that the randomized allocation gives us meaningful PROP1 guarantees with high probability against non-adaptive adversaries; (ii) lightweight predictions such as estimates of maximum item values can admit robust PROP1 approximations against adaptive adversaries. These results position PROP1 as a fairness notion that is both theoretically informative and algorithmically tractable in online settings, relative to EF1, MMS, and PROPX.

Several questions remain open. Most notably, can one obtain any nontrivial approximation to PROP1 *without* predictive or structural assumptions? Beyond known impossibility results, no approximation bounds are currently established in this setting, and resolving this remains a key challenge.

---

[8]Our code is available at https://github.com/nicteh/Approx-Prop-Online-Fair-Division.

## Acknowledgments

Nicholas Teh was supported by the Advanced Research + Invention Agency (ARIA) as part of the ASPAI project.

## Impact Statement

This paper presents work whose goal is to advance the theory of machine learning. There are many potential societal consequences of our work, none of which we feel must be specifically highlighted here.

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

# Appendix

## A. Further Related Work

**Learning-Augmented Algorithms.** Since the seminal work of Lykouris & Vassilvitskii (2021), there has been a surge of interest in incorporating unreliable advice into algorithm design and analyzing performance as a function of advice quality across various areas of computer science. This framework has been especially successful in online optimization, where the core challenge lies in handling uncertainty about future inputs. In this context, advice can serve as a useful proxy for the unknown future. Most previous works in this setting are in the context of online algorithms, e.g. for the ski-rental problem (Gollapudi & Panigrahi, 2019; Wang et al., 2020; Angelopoulos et al., 2024), non-clairvoyant scheduling (Purohit et al., 2018), scheduling (Lattanzi et al., 2020; Bamas et al., 2020a; Antoniadis et al., 2022), augmenting classical data structures with predictions (e.g. indexing (Kraska et al., 2018) and Bloom filters (Mitzenmacher, 2018)), online selection and matching problems (Antoniadis et al., 2023; Dütting et al., 2021; Choo et al., 2024; 2025), online TSP (Bernardini et al., 2022; Gouleakis et al., 2023), and a more general framework of online primal-dual algorithms (Bamas et al., 2020b). However, there have been some recent applications to other areas, e.g. graph algorithms (Chen et al., 2022; Dinitz et al., 2021), causal learning (Choo et al., 2023), mechanism design (Gkatzelis et al., 2022; Agrawal et al., 2022), distribution learning (Bhattacharyya et al., 2025a;b), and approval elicitation (Choo et al., 2026b). For an overview of this growing area, we refer the reader to the survey by Mitzenmacher & Vassilvitskii (2022).[9]

**Online Voting.** Beyond fair division, online and temporal models have also been studied in voting, which can be viewed as a *public goods* setting since the selected outcomes are shared by all voters. For instance, the multiwinner voting literature contains a growing body of work on online and temporal candidate or committee selection (Lackner, 2020; Alouf-Heffetz et al., 2022; Do et al., 2022; Elkind et al., 2022; Lackner & Maly, 2023; Chandak et al., 2024; Elkind et al., 2024a;b; Zech et al., 2024; Elkind et al., 2025b;c; Phillips et al., 2026; Teh, 2026).

## B. Omitted Proofs from Section 3

### B.1. Proof of Proposition 3.3

**Lemma B.1.** *Suppose goods have been allocated up to and including timestep $t \in \mathbb{N}_+$. Assume that $v_j(G^{(t)}) > 0$ and $c_j := c_j^{(t)} > 0$ for all $j \in [n]$, and that there is some $i \in [n]$ such that $\alpha_i^{(t)} < \alpha_j^{(t)}$ for all $j \neq i$. Then, for any nonempty subset $S \subseteq [n]$ not containing $i$, there exist a nonnegative integer $\tau$ and a sequence of $\tau$ goods $g_{t+1}, g_{t+2}, \cdots, g_{t+\tau}$ such that* Greedy Strategy 3 *must give:*

- $c_j^{(t+\tau)} = c_j^{(t)}$ and $v_j(A_j^{(t+\tau)}) = v_j(A_j^{(t)})$ for all $j \in [n]$,

- $v_j(G^{(t+\tau)}) = v_j(G^{(t)})$ and $\alpha_j^{(t+\tau)} = \alpha_j^{(t)}$ for $j \in [n] \setminus S$, and

- $v_j(G^{(t+\tau)}) = v_j(G^{(t)}) + \frac{c_j}{2}\tau_j$ and $\alpha_i^{(t)} < \alpha_j^{(t+\tau)} \leq \alpha_i^{(t)}(1+\delta_j^{(t+\tau)})$ for $j \in S$, where $\tau_j := \lceil \frac{2}{c_j} v_j(G^{(t)})(\alpha_j^{(t)}/\alpha_i^{(t)} - 1) \rceil - 1$ and $\delta_j^{(t')} := c_j/(2v_j(G^{(t')}))$ for $t' \geq t$.

*In fact, we may take $\tau = \max_{j \in S} \tau_j$.*

*Proof.* For all $j$ and all $t' \geq t$, we inductively define the good $g_{t'+1}$ to have valuation

$$v_j(g_{t'+1}) = \begin{cases} c_j/2 & \text{if } j \in S \text{ and } \alpha_j^{(t')} > \alpha_i^{(t)}(1+\delta_j^{(t')}), \\ 0 & \text{otherwise.} \end{cases}$$

First, notice that $v_j(g_{t'+1}) < c_j = c_j^{(t)}$ for all $t' \geq t$, so $c_j^{(t')} = c_j$ by definition. Also, because $v_j(g_{t'+1}) = 0$ for all $t' \geq t$ and all $j \in [n] \setminus S$, it follows that $v_j(A_j^{(t')}) = v_j(A_j^{(t)})$ and $v_i(G^{(t')}) = v_i(G^{(t)})$ and $\alpha_j^{(t')} = \alpha_j^{(t)}$ for all such $t', j$.

---

[9]See also https://algorithms-with-predictions.github.io/.

Moreover, observe that if $\alpha_j^{(t')} \leq \alpha_i^{(t)}(1 + \delta_j^{(t')})$ for some $j \in S$ and $t' \geq t$, then $\alpha_j^{(t'')} \leq \alpha_i^{(t)}(1 + \delta_j^{(t'')})$ for *all* $t'' \geq t'$ by induction on $t''$, using that $v_j(g_{t''+1}) = 0$ implies $\alpha_j^{(t''+1)} = \alpha_j^{(t'')}$ and $\delta_j^{(t''+1)} = \delta_j^{(t'')}$ by definition.

For this sequence of goods, we will show that whenever $t' \geq t$, the statement $P(t')$ that $\alpha_i^{(t)} < \alpha_j^{(t')}$ for $j \neq i$ and the statement $Q(t')$ that the good $g_{t'+1}$ is given to agent $i$ are always true. Note that $P(t)$ holds by assumption.

The condition for $Q(t')$ to hold, i.e., for $g_{t'+1}$ to be allocated to agent $i$, is:

$$\frac{v_i(A_i^{(t')}) + c_i}{v_i(G^{(t')}) + v_i(g_{t'+1})} < \frac{v_j(A_j^{(t')}) + c_j}{v_j(G^{(t')}) + v_j(g_{t'+1})} \tag{1}$$

for all $j \neq i$. Noting that $v_i(g_{t'+1}) = 0$, this is equivalent to

$$\frac{v_i(A_i^{(t')}) + c_i}{v_i(G^{(t')})} < \frac{v_j(A_j^{(t')}) + c_j}{v_j(G^{(t')})} \frac{v_j(G^{(t')})}{v_j(G^{(t')}) + v_j(g_{t'+1})},$$

i.e., $\alpha_i^{(t')} < \alpha_j^{(t')}/(1 + v_j(g_{t'+1})/v_j(G^{(t')}))$ for all $j \neq i$. If $j \neq i$ is such that $v_j(g_{t'+1}) = 0$, then this simplifies to $\alpha_i^{(t)} < \alpha_j^{(t')}$ (given the observation that $\alpha_i^{(t)} = \alpha_i^{(t')}$), which will be true if $P(t')$ is. If instead $j \neq i$ is such that $v_j(g_{t'+1}) = c_j/2$ (i.e., $j \in S$ and $\alpha_j^{(t')} > \alpha_i^{(t)}(1 + \delta_j^{(t')})$), then the inequality instead simplifies to the true statement $\alpha_j^{(t')} > \alpha_i^{(t)}(1 + c_j/(2v_j(G^{(t')})))$ (noting $c_j/(2v_j(G^{(t')})) = \delta_j^{(t')}$). Thus $Q(t')$ follows from $P(t')$.

Assuming $Q(t')$ holds, it means the inequality in (1) holds for all $j \neq i$. As noted earlier, the LHS is simply $\alpha_i^{(t)}$, while the RHS is in fact $\alpha_j^{(t'+1)}$ since $g_{t'+1}$ was indeed not allocated to agent $j$. Hence $P(t'+1)$ holds.

Thus, $P(t')$ and $Q(t')$ always hold for all $t' \geq t$. In particular, the goods are always allocated to agent $i$ and we have $v_j(A_j^{(t')}) = v_j(A_j^{(t)})$ for all $j \in [n]$ (noting all goods have value 0 to agent $i$).

Fix $j \in S$. As noted earlier, for any $t' \geq t$, if $\alpha_j^{(t')} \leq \alpha_i^{(t)}(1 + \delta_j^{(t')})$ holds, then the same holds when we replace $t'$ by any $t'' \geq t'$. Let $\tau_j'$ be the smallest nonnegative integer (or $\infty$ if none exists) such that $\alpha_j^{(t+\tau_j')} \leq \alpha_i^{(t)}(1 + \delta_j^{(t+\tau_j')})$, so $\alpha_j^{(t')} > \alpha_i^{(t)}(1 + \delta_j^{(t')})$ is true precisely when $t \leq t' < t + \tau_j'$. This means $v_j(g_{t'+1}) = c_j/2$ for $t' \in \{t, t+1, \cdots, t+\tau_j'-1\}$[10] and $v_j(g_{t'+1}) = 0$ otherwise. Thus, for any (finite) nonnegative integer $\sigma$, if we define $\sigma_j := \min(\tau_j', \sigma)$, then the statement $\alpha_j^{(t+\sigma)} > \alpha_i^{(t)}(1 + \delta_j^{(t+\sigma)})$ is equivalent to

$$\frac{v_j(A_j^{(t)}) + c_j}{v_j(G^{(t)}) + \sigma_j c_j/2} > \alpha_i^{(t)} \left(1 + \frac{c_j}{2(v_j(G^{(t)}) + \sigma_j c_j/2)}\right).$$

Multiplying both sides by $v_j(G^{(t)}) + \sigma_j c_j/2$ and using $v_j(A_j^{(t)}) + c_j = \alpha_j^{(t)} v_j(G^{(t)})$ gives the equivalent statement

$$\alpha_j^{(t)} v_j(G^{(t)}) > \alpha_i^{(t)}(v_j(G^{(t)}) + \sigma_j c_j/2 + c_j/2),$$

which simplifies to the equivalent

$$\sigma_j < \tfrac{2}{c_j} v_j(G^{(t)})(\alpha_j^{(t)}/\alpha_i^{(t)} - 1) - 1.$$

But the statement $\alpha_j^{(t+\sigma)} > \alpha_i^{(t)}(1 + \delta_j^{(t+\sigma)})$ is also equivalent to $t \leq t + \sigma < t + \tau_j'$, i.e., to $\sigma < \tau_j'$, which means we have the equivalence

$$\sigma < \tau_j' \iff \min(\tau_j', \sigma) < \tfrac{2}{c_j} v_j(G^{(t)})(\alpha_j^{(t)}/\alpha_i^{(t)} - 1) - 1.$$

In particular, this means $\tau_j' \neq \infty$, because otherwise the left side of the equivalence will always hold but the right side of the equivalence will fail for sufficiently large $\sigma$. Moreover, taking $\sigma = \tau_j'$ gives

$$\tau_j' \geq \tfrac{2}{c_j} v_j(G^{(t)})(\alpha_j^{(t)}/\alpha_i^{(t)} - 1) - 1.$$

---

[10]The set $\{t, t+1, \cdots, t+\tau_j'-1\}$ is the empty set when $\tau_j = 0$ and is the infinite set $\{t, t+1, t+2, \cdots\}$ when $\tau_j = \infty$.

If $\tau'_j > 0$, we may take $\sigma = \tau'_j - 1$ to get

$$\tau'_j - 1 < \tfrac{2}{c_j} v_j(G^{(t)})(\alpha_j^{(t)}/\alpha_i^{(t)} - 1) - 1,$$

from which we obtain

$$\tau'_j = \lceil \tfrac{2}{c_j} v_j(G^{(t)})(\alpha_j^{(t)}/\alpha_i^{(t)} - 1) \rceil - 1 =: \tau_j.$$

If $\tau'_j = 0$, this last formula is still true, because the earlier inequality derived from taking $\sigma = 0$ has already shown $\tfrac{2}{c_j} v_j(G^{(t)})(\alpha_j^{(t)}/\alpha_i^{(t)} - 1) \le 1$, but this quantity is clearly strictly positive since $\alpha_j^{(t)} > \alpha_i^{(t)}$.

Thus, for any $j \in S$, we have $\alpha_j^{(t')} \le \alpha_i^{(t)}(1 + \delta_j^{(t')})$ and $v_j(G^{(t')}) = v_j(G^{(t)}) + \tfrac{c_j}{2}\tau_j$ whenever $t' \ge t + \tau_j$. As such, the lemma holds if we take $\tau = \max_{j \in S} \tau_j$ and goods $g_{t+1}, \cdots, g_{t+\tau}$ as defined at the beginning of the proof. $\square$

**Proposition B.2.** *For $n \ge 2$ and any $\alpha > 0$, there exists a sequence of $m = m(n, \alpha)$ arriving goods such that* Greedy Strategy 3 *fails to produce an $\alpha$-PROP1 allocation.*

*Proof.* It suffices to prove that there is an infinite sequence of goods $g_1, g_2, \cdots$ such that $\liminf_{t \to \infty} \alpha^{(t)} = 0$, where $\alpha^{(t)} := \min_{i \in [n]} \alpha_i^{(t)}$.

We begin by setting $v_i(g_1) = 1$ for all $i \in [n]$. Without loss of generality, suppose the good $g_1$ is given to agent 1. Next, for $t \in \{2, 3\}$, set $v_i(g_t) = 1$ for $i \in \{1, 2\}$ and $v_i(g_2) = 0$ otherwise. A straightforward calculation shows that the good $g_2$ must be given to agent 2, and that $g_3$ must be given to either agent 1 or 2. Without loss of generality, suppose $g_3$ is given to agent 1. Under this setup, $1 = \alpha_1^{(3)} > \alpha_2^{(3)} = 2/3$, with $v_1(A_1^{(3)}) = 2$, $v_2(A_2^{(3)}) = 1$, $v_1(G^{(3)}) = v_2(G^{(3)}) = 3$ and $c_1^{(3)} = c_2^{(3)} = 1$. Also, $v_i(A_i^{(3)}) = 0$ and $\alpha_i^{(3)} = c_i^{(3)} = v_i(G^{(3)}) = 1$ for all $i \ge 3$.

We now inductively define the goods $g_4, g_5, \cdots$ and an infinite sequence of time steps $3 = t_0 < t_1 < t_2 < \cdots$ such that for each nonnegative integer $k$,

1. $v_r(A_r^{(t_k)}) = v_r(A_r^{(t_0)})$, $c_r^{(t_k)} = c_r^{(t_0)}$, $v_r(G^{(t_k)}) = v_r(G^{(t_0)})$, and $\alpha_r^{(t_k)} = \alpha_r^{(t_0)}$ for all $r \ge 3$,

2. $v_r(A_r^{(t_k)}) \le v_r(A_r^{(t_0)}) + kc_r^{(t_0)}$ and $c_r^{(t_k)} = c_r^{(t_0)}$ for all $r \in \{1, 2\}$, and

3. there is some $i \in \{1, 2\}$ such that $\alpha_i^{(t_k)} < \alpha_r^{(t_k)}$ for all $r \ne i$. (So $\alpha^{(t_k)} = \alpha_i^{(t_k)}$.)

These conditions are met for $k = 0$, with $i = 2$.

For $k \ge 0$, assume that we have defined $t_0, \cdots, t_k$ and goods up to and including $g_{t_k}$. The conditions of Lemma B.1 are met at time step $t_k$, so taking $S = \{3 - i\}$[11] in the lemma produces a sequence of

$$\tau_k := \lceil \tfrac{2}{c_{3-i}^{(t_k)}} v_{3-i}(G^{(t_k)})(\alpha_{3-i}^{(t_k)}/\alpha_i^{(t_k)} - 1) \rceil - 1 \ge 0$$

goods $g_{t_k+1}, g_{t_k+2}, \cdots, g_{t_k+\tau_k}$ such that:

- $c_r^{(t_k+\tau_k)} = c_r^{(t_k)} = c_r^{(t_0)}$ and $v_r(A_r^{(t_k+\tau_k)}) = v_r(A_r^{(t_k)})$ for all $r \in [n]$,

- $v_r(G^{(t_k+\tau_k)}) = v_r(G^{(t_k)})$ and $\alpha_r^{(t_k+\tau_k)} = \alpha_r^{(t_k)}$ for $r \ne 3 - i$, and

- $v_{3-i}(G^{(t_k+\tau_k)}) = v_{3-i}(G^{(t_k)}) + \tfrac{c_{3-i}^{(t_k)}}{2}\tau_k$ and $\alpha_i^{(t_k)} < \alpha_{3-i}^{(t_k+\tau_k)} \le \alpha_i^{(t_k)}(1 + \delta_{3-i}^{(t_k+\tau_k)})$, where $\delta_{3-i}^{(t_k+\tau_k)} := c_{3-i}^{(t_k)}/(2v_{3-i}(G^{(t_k+\tau_k)}))$.

---

[11] Note that $3 - i$ is the element in $\{1, 2\}$ different from $i$.

Observe that

$$
\begin{aligned}
v_{3-i}(G^{(t_k+\tau_k)}) &= v_{3-i}(G^{(t_k)}) + \frac{c_{3-i}^{(t_k)}}{2}\tau_k \\
&\geq v_{3-i}(G^{(t_k)}) + \frac{c_{3-i}^{(t_k)}}{2}\left( \frac{2v_{3-i}(G^{(t_k)})}{c_{3-i}^{(t_k)}}\left( \frac{\alpha_{3-i}^{(t_k)}}{\alpha_i^{(t_k)}} - 1 \right) - 1 \right) \\
&= \frac{v_{3-i}(G^{(t_k)})\alpha_{3-i}^{(t_k)}}{\alpha_i^{(t_k)}} - \frac{c_{3-i}^{(t_k)}}{2} \\
&= \frac{v_{3-i}(A_{3-i}^{(t_k)}) + c_{3-i}^{(t_0)}}{\alpha_i^{(t_k)}} - \frac{c_{3-i}^{(t_0)}}{2},
\end{aligned}
$$

and thus

$$
v_r(G^{(t_k+\tau_k)}) \geq \frac{v_r(A_r^{(t_k)}) + c_r^{(t_0)}}{\alpha_i^{(t_k)}} - \frac{c_r^{(t_0)}}{2} \tag{2}
$$

for both $r \in \{1, 2\}$, noting that this is true for $r = i$ since the left-hand side is equal to the first term on the right-hand side.

We can now define good $g_{t_k+\tau_k+1}$ to be of value $c_r^{(t_0)}$ to each agent $r \in \{1, 2\}$ and value 0 to the other agents. The greedy strategy must allocate the good to either agent 1 or 2, because the value of $\alpha_r^{(t_k+\tau_k+1)}$ should agent $r$ not be allocated the good will be $\alpha_r^{(t_k+\tau_k)} = \alpha_r^{(t_k)} > \alpha_i^{(t_k)}$ for $r \geq 3$ but will be less than $\alpha_i^{(t_k+\tau_k)} = \alpha_i^{(t_k)}$ for $r = i$ (because the nonzero numerator in the calculation of the ratio remains the same but the denominator strictly increases).

If the greedy strategy allocates $g_{t_k+\tau_k+1}$ to agent $3 - i$ (rather than agent $i$), then we have

$$
\alpha_i^{(t_k+\tau_k+1)} < \alpha_i^{(t_k+\tau_k)} = \alpha_i^{(t_k)} < \alpha_{3-i}^{(t_k+\tau_k)} \leq \alpha_{3-i}^{(t_k+\tau_k+1)},
$$

giving us $\alpha^{(t_k+\tau_k+1)} = \alpha_i^{(t_k+\tau_k+1)} < \alpha^{(t_k)}$. If instead the greedy algorithm allocates $g_{t_k+\tau_k+1}$ to agent $i$ (rather than agent $3 - i$), we will have

$$
\begin{aligned}
\alpha_{3-i}^{(t_k+\tau_k+1)} &= \frac{v_{3-i}(A_{3-i}^{(t_k+\tau_k)}) + c_{3-i}^{(t_k+\tau_k+1)}}{v_{3-i}(G^{(t_k+\tau_k)}) + c_{3-i}^{(t_k+\tau_k+1)}} \\
&= \frac{v_{3-i}(A_{3-i}^{(t_k+\tau_k)}) + c_{3-i}^{(t_0)}}{v_{3-i}(G^{(t_k+\tau_k)}) + c_{3-i}^{(t_0)}} \\
&= \alpha_{3-i}^{(t_k+\tau_k)}\frac{v_{3-i}(G^{(t_k+\tau_k)})}{v_{3-i}(G^{(t_k+\tau_k)}) + c_{3-i}^{(t_0)}} \\
&\leq \alpha_i^{(t_k)}(1 + \delta_{3-i}^{(t_k+\tau_k)})\frac{v_{3-i}(G^{(t_k+\tau_k)})}{v_{3-i}(G^{(t_k+\tau_k)}) + c_{3-i}^{(t_0)}} \\
&= \alpha_i^{(t_k)}\left( 1 + \frac{c_{3-i}^{(t_0)}}{2v_{3-i}(G^{(t_k+\tau_k)})} \right)\frac{v_{3-i}(G^{(t_k+\tau_k)})}{v_{3-i}(G^{(t_k+\tau_k)}) + c_{3-i}^{(t_0)}} \\
&< \alpha_i^{(t_k)}\left( 1 + \frac{c_{3-i}^{(t_0)}}{v_{3-i}(G^{(t_k+\tau_k)})} \right)\frac{v_{3-i}(G^{(t_k+\tau_k)})}{v_{3-i}(G^{(t_k+\tau_k)}) + c_{3-i}^{(t_0)}} \\
&= \alpha_i^{(t_k)} = \alpha_i^{(t_k+\tau_k)} \leq \alpha_i^{(t_k+\tau+1)},
\end{aligned}
$$

giving us $\alpha^{(t_k+\tau_k+1)} = \alpha_{3-i}^{(t_k+\tau_k+1)} < \alpha^{(t_k)}$. Note that by the inequality in (2),

$$
v_r(G^{(t_k+\tau_k+1)}) = v_r(G^{(t_k+\tau_k)}) + c_r^{(t_0)} \geq \frac{v_r(A_r^{(t_k)}) + c_r^{(t_0)}}{\alpha_i^{(t_k)}} + \frac{c_r^{(t_0)}}{2} \tag{3}
$$

for $r \in \{1, 2\}$.

Now take $t_{k+1} = t_k + \tau_k + 1 > t_k$. We need to show that the three numbered conditions imposed earlier continue to hold for $k$ replaced with $k + 1$. The first condition regarding the values of $v_r(A_r^{(t_{k+1})}), v_r(G^{(t_{k+1})}), c_r^{(t_{k+1})}, \alpha_r^{(t_{k+1})}$ when $r \geq 3$ can easily be verified since only goods of value 0 are given to these agents. We just showed that the third condition holds, with the new value of $i$ being the agent (among 1 and 2) that did not receive good $g_{t_k+\tau_k+1}$. Finally, the second condition

follows by induction, noting that none of the new goods have value to agent $r \in \{1, 2\}$ exceeding $c_r^{(t_0)}$ and that the only allocated good of nonzero value is good $g_{t_k + \tau_k + 1}$ of value $c_r^{(t_0)}$ to agent $r \in \{1, 2\}$.

This completes the construction of the infinite sequence of goods $g_1, g_2, g_3, \cdots$.

We will now show that $\liminf_{t \to \infty} \alpha^{(t)} = 0$. To this end, it suffices to show that $\lim_{k \to \infty} (1/\alpha^{(t_k)}) = \infty$, for then $\lim_{k \to \infty} \alpha^{(t_k)} = 0$.

Fix $k \geq 0$. Suppose that $i, j \in \{1, 2\}$ satisfy $\alpha^{(t_k)} = \alpha_i^{(t_k)} < \alpha_{3-i}^{(t_k)}$ and $\alpha^{(t_{k+1})} = \alpha_j^{(t_{k+1})} < \alpha_{3-j}^{(t_{k+1})}$. Then

$$
\begin{aligned}
\frac{1}{\alpha^{(t_{k+1})}} &= \frac{v_j(G^{(t_{k+1})})}{v_j(A_j^{(t_{k+1})}) + c_j^{(t_0)}} \\
&\geq \frac{\left( \frac{v_j(A_j^{(t_k)}) + c_j^{(t_0)}}{\alpha_i^{(t_k)}} + \frac{c_j^{(t_0)}}{2} \right)}{v_j(A_j^{(t_k)}) + c_j^{(t_0)}} \\
&= \frac{1}{\alpha^{(t_k)}} + \frac{c_j^{(t_0)}/2}{v_j(A_j^{(t_k)}) + c_j^{(t_0)}} \\
&\geq \frac{1}{\alpha^{(t_k)}} + \frac{c_j^{(t_0)}/2}{v_j(A_j^{(t_0)}) + k c_j^{(t_0)} + c_j^{(t_0)}} \\
&= \frac{1}{\alpha^{(t_k)}} + \frac{1}{2(v_j(A_j^{(t_0)})/c_j^{(t_0)} + k + 1)}
\end{aligned}
$$

where we applied the inequality in (3) at the second step. Taking $\lambda := \max(v_1(A_1^{(t_0)})/c_1^{(t_0)}, v_2(A_2^{(t_0)})/c_2^{(t_0)})$ (which is 2 in our case), we see that

$$
\frac{1}{\alpha^{(t_{k+1})}} - \frac{1}{\alpha^{(t_k)}} \geq \frac{1}{2(k + 1 + \lambda)}.
$$

The above holds for any $k \geq 0$, so a telescoping sum of these inequalities gives

$$
\frac{1}{\alpha^{(t_k)}} \geq \frac{1}{\alpha^{(t_0)}} + \sum_{s=1}^{k} \frac{1}{2(s + \lambda)}.
$$

The right-hand side tends to $\infty$ as $k \to \infty$, so $\lim_{k \to \infty} (1/\alpha^{(t_k)}) = \infty$, as desired. $\qquad \square$

## C. Omitted Proofs from Section 4

### C.1. Proof of Theorem 4.1

Fix an agent $i \in [n]$. Let $\alpha := \frac{3}{32 \log(n/\delta)}$. We will first show that for $g := \arg\max_{g' \in G \setminus A_i} v_i(g)$,

$$
\Pr\left[ v_i(A_i \cup \{g\}) < \alpha \cdot \frac{v_i(G)}{n} \right] \leq \frac{\delta}{n}. \tag{4}
$$

Once the above is proved, a union bound over all $n$ agents will imply that the probability the allocation is $\alpha$-PROP1 is at least $1 - \delta$.

Then, since there exists a $g \in G \setminus A_i$ such that $v_i(A_i \cup \{g\}) \geq \max_{j \in [m]} v_i(g_j)$, if $\max_{j \in [m]} v_i(g_j) \geq \alpha \cdot \frac{v_i(G)}{n}$, we get that

$$
v_i(A_i \cup \{g\}) \geq \max_{j \in [m]} v_i(g_j) \geq \alpha \cdot \frac{v_i(G)}{n}
$$

and equivalently, $\Pr\left[ v_i(A_i \cup \{g\}) < \alpha \cdot \frac{v_i(G)}{n} \right] = 0$, giving us (4) as desired. Thus, it suffices to consider the case when $\max_{j \in [m]} v_i(g_j) < \alpha \cdot \frac{v_i(G)}{n}$.

Consider any agent $i \in [n]$. Define $X_j := \mathbf{1}\{g_j \in A_i\}$ and $X_j' := 1 - X_j = \mathbf{1}\{g_j \notin A_i\}$. Then, we have that $X_1, \ldots, X_m \sim \text{Bernoulli}(1/n)$ and $X_1', \ldots, X_m' \sim \text{Bernoulli}\left(\frac{n-1}{n}\right)$. Let $S_i = \sum_{j \in [m]} (v_i(g_j) \cdot X_j)$ be a random variable

representing $v_i(A_i)$; and let $S'_i = \sum_{j \in [m]} (v_i(g_j) \cdot X'_j)$. Now, for each $i \in [n]$, $S_i = v_i(G) - S'_i$,

$$
\Pr\left[v_i(A_i \cup \{g\}) < \frac{\alpha \cdot v_i(G)}{n}\right] \leq \Pr\left[v_i(A_i) < \frac{\alpha \cdot v_i(G)}{n}\right]
$$

$$
= \Pr\left[S_i < \frac{\alpha \cdot v_i(G)}{n}\right]
$$

$$
= \Pr\left[S'_i > \frac{(n-1) \cdot v_i(G)}{n} + \frac{(1-\alpha) \cdot v_i(G)}{n}\right]
$$

We now prove several properties of the random variable $S'_i$, relating to its expectation, variance, and the individual terms $v_i(g_j) \cdot X'_j$ in the summation.

**Lemma C.1.** *For $S'_i = \sum_{j \in [m]} (v_i(g_j) \cdot X'_j)$, the following properties always hold:*

*(i) $\mathbb{E}[S'_i] = \frac{(n-1)}{n} \cdot v_i(G)$;*

*(ii) $\sigma^2(S'_i) \leq \frac{\alpha \cdot v_i(G)^2}{n^2}$; and*

*(iii) $v_i(g_j) \cdot X'_j - \mathbb{E}[v_i(g_j) \cdot X'_j] \leq \frac{\alpha \cdot v_i(G)}{n}$ for all $j \in [m]$.*

*Proof.* We first prove (i). By linearity of expectation, we get that

$$
\mathbb{E}[S'_i] = \sum_{j \in [m]} (v_i(g_j) \cdot \mathbb{E}[X'_j]) = \sum_{j \in [m]} v_i(g_j) \cdot \frac{n-1}{n} = \frac{n-1}{n} \cdot v_i(G),
$$

where the last line follows from the fact that $\sum_{j \in [m]} v_i(g_j) = v_i(G)$. Next, we prove (ii). Since $v_i(g_j) \leq \alpha \cdot \frac{v_i(G)}{n}$ for all $j \in [m]$, we have that

$$
\sigma^2(S'_i) = \sum_{j \in [m]} (v_i(g_j) \cdot v_i(g_j) \cdot \sigma^2(X'_j)) \leq \sum_{j \in [m]} (v_i(g_j) \cdot \alpha \cdot v_i(G) \cdot \sigma^2(X'_j))
$$

$$
= \sum_{j \in [m]} \left(v_i(g_j) \cdot \alpha \cdot \frac{v_i(G)}{n} \cdot \frac{n-1}{n^2}\right)
$$

$$
= \alpha \cdot \frac{v_i(G)}{n} \cdot \frac{n-1}{n^2} \cdot \sum_{j \in [m]} v_i(g_j)
$$

$$
= \alpha \cdot \frac{v_i(G)}{n} \cdot \frac{n-1}{n^2} \cdot v_i(G)
$$

$$
\leq \frac{\alpha \cdot v_i(G)^2}{n^2}.
$$

Finally, we prove (iii). For all $j \in [m]$, since $X'_j \leq 1$ and $v_i(g) \leq \alpha \cdot \frac{v_i(G)}{n}$, we have that

$$
v_i(g_j) \cdot X'_j - \mathbb{E}[v_i(g_j) \cdot X'_j] \leq v_i(g_j) \cdot \left(1 - \frac{n-1}{n}\right) = \frac{v_i(g_j)}{n} \leq v_i(g_j) \leq \alpha \cdot \frac{v_i(G)}{n}. \qquad \square
$$

We now state Bernstein's inequality, which we will use for the proof.

**Lemma C.2** (Dubhashi & Panconesi (2009); Theorem 1.4)**.** *Let the random variables $X_1, \ldots, X_n$ be independent with $X_i - \mathbb{E}[X_i] \leq b$ for each $i \in [n]$. Also let $X := \sum_{i \in [n]} X_i$ and $\sigma^2 := \sum_{i \in [n]} \sigma_i^2$ be the variance of $X$. Then, for any $t > 0$,*

$$
\Pr[X > \mathbb{E}[X] + t] \leq \exp\left(-\frac{t^2}{2\sigma^2 + 2bt/3}\right).
$$

We can then apply Bernstein's inequality to $S_i'$. By setting $t = \frac{(1-\alpha) \cdot v_i(G)}{n}$ and $b = \alpha \cdot \frac{v_i(G)}{n}$, we get

$$\Pr\left[S_i' > \frac{(n-1) \cdot v_i(G)}{n} + \frac{(1-\alpha) \cdot v_i(G)}{n}\right] = \Pr\left[S_i' > \mathbb{E}[S'] + t\right] \leq \exp\left(-\frac{t^2}{2\sigma^2 + 2bt/3}\right),$$

where the equality follows from property (i) of Lemma C.1. We also note that Bernstein's inequality is applicable due to property (iii) of Lemma C.1.

Since our goal is to find an upper bound to our expression, it suffices to use a lower bound on the numerator $t^2$, and an upper bound for the denominator $2\sigma^2 + 2bt/3$.

For the numerator $t^2$, since $\alpha = \frac{3}{32 \log (n/\delta)} < \frac{1}{2}$ for all $n \geq 2$, we have that

$$t^2 = \frac{(1-\alpha)^2 \cdot v_i(G)^2}{n^2} \geq \frac{v_i(G)^2}{4n^2}$$

For the denominator $2\sigma^2 + 2bt/3$, by bounding our variance with property (ii) of Lemma C.1, we have that

$$2\sigma^2 + \frac{2bt}{3} \leq \frac{2\alpha \cdot v_i(G)^2}{n^2} + \frac{2\alpha(1-\alpha) \cdot v_i(G)^2}{3n^2} < \frac{2\alpha \cdot v_i(G)^2}{n^2} + \frac{2\alpha \cdot v_i(G)^2}{3n^2} = \frac{8\alpha \cdot v_i(G)^2}{3n^2},$$

where the second inequality follows from the fact that $\alpha > 0$ and hence $1 - \alpha < 1$. Thus, we get that

$$\exp\left(-\frac{t^2}{2\sigma^2 + 2bt/3}\right) \leq \exp\left(-\frac{\frac{v_i(G)^2}{4n^2}}{\frac{8\alpha \cdot v_i(G)^2}{3n^2}}\right) = \exp\left(-\frac{3}{32 \cdot \alpha}\right) = \exp\left(-\ln(n/\delta)\right) = \frac{\delta}{n},$$

giving us (4). Then, by summing the bound in (4) over the $n$ agents: the probability that any agent violates the $\alpha$-PROP1 condition is at most $n \cdot \frac{\delta}{n} = \delta$. Therefore, with probability at least $1 - \delta$, for every agent $i \in [n]$ and $g := \max_{g' \in G \setminus A_i} v_i(g')$,

$$v_i(A_i \cup \{g\}) \geq \alpha \cdot \frac{v_i(G)}{n},$$

and the allocation returned is $\alpha$-PROP1.

## C.2. Proof of Theorem 4.2

Let

$$m := \left\lceil \frac{\log\left(\frac{n}{2\delta}\right)}{\log\left(\frac{n}{n-1}\right)} \right\rceil \quad \text{and} \quad \alpha := \frac{2n}{m}. \tag{5}$$

Consider the instance with $m$ goods $G = \{g_1, \ldots, g_m\}$ where all agents have identical additive valuations

$$v_i(g_j) = 1 \text{ for all } i \in [n] \text{ and } j \in [m].$$

Assume a non-adaptive adversary. Run RAND, which allocates each good independently and uniformly at random to one of the $n$ agents. Let $\mathcal{A} = (A_1, \ldots, A_n)$ denote the (random) final allocation. For each agent $i \in [n]$, define the event

$$E_i := \{A_i = \varnothing\},$$

i.e., agent $i$ receives no goods. For a fixed agent $i$,

$$\Pr[E_i] = \left(1 - \frac{1}{n}\right)^m,$$

and for distinct $i \neq j$,

$$\Pr[E_i \cap E_j] = \left(1 - \frac{2}{n}\right)^m.$$

Let $p_1 := (1 - \frac{1}{n})^m$ and $p_2 := (1 - \frac{2}{n})^m$. By the first two terms of Bonferroni inequalities,

$$\Pr\left[\bigcup_{i=1}^n E_i\right] \geq \sum_{i=1}^n \Pr[E_i] - \sum_{1 \leq i < j \leq n} \Pr[E_i \cap E_j] = np_1 - \binom{n}{2}p_2. \tag{6}$$

Moreover, since $(1 - \frac{1}{n})^2 = 1 - \frac{2}{n} + \frac{1}{n^2} \geq 1 - \frac{2}{n}$, we have $p_2 \leq p_1^2$. Plugging this into (6) gives us

$$\Pr\left[\bigcup_{i=1}^n E_i\right] \geq np_1 - \binom{n}{2}p_1^2 = np_1\left(1 - \frac{(n-1)p_1}{2}\right). \tag{7}$$

We now lower bound $p_1$. Define

$$m_0 := \frac{\log\left(\frac{n}{2\delta}\right)}{\log\left(\frac{n}{n-1}\right)}.$$

By definition, $m = \lfloor m_0 \rfloor$, so $m \leq m_0 < m + 1$. Since $1 - \frac{1}{n} \in (0,1)$, the function $f(x) = (1 - \frac{1}{n})^x$ is decreasing. Thus,

$$p_1 = \left(1 - \frac{1}{n}\right)^m \geq \left(1 - \frac{1}{n}\right)^{m_0}.$$

Using $1 - \frac{1}{n} = \frac{n-1}{n} = \left(\frac{n}{n-1}\right)^{-1}$, we get

$$\left(1 - \frac{1}{n}\right)^{m_0} = \left(\frac{n}{n-1}\right)^{-m_0} = \exp\left(-\log\left(\frac{n}{n-1}\right) \cdot \frac{\log(\frac{n}{2\delta})}{\log(\frac{n}{n-1})}\right) = \exp\left(-\log\left(\frac{n}{2\delta}\right)\right) = \frac{2\delta}{n}.$$

Therefore,

$$p_1 \geq \frac{2\delta}{n}. \tag{8}$$

Similarly, since $m_0 < m + 1$,

$$\left(1 - \frac{1}{n}\right)^{m+1} < \left(1 - \frac{1}{n}\right)^{m_0} = \frac{2\delta}{n},$$

and thus

$$p_1 = \left(1 - \frac{1}{n}\right)^m = \frac{\left(1 - \frac{1}{n}\right)^{m+1}}{1 - \frac{1}{n}} < \frac{\frac{2\delta}{n}}{\frac{n-1}{n}} = \frac{2\delta}{n-1}. \tag{9}$$

Plugging (8) and (9) into (7), we get

$$\Pr\left[\bigcup_{i=1}^n E_i\right] \geq n \cdot \frac{2\delta}{n}\left(1 - \frac{(n-1)}{2} \cdot \frac{2\delta}{n-1}\right) = 2\delta(1 - \delta) \geq \delta,$$

where the last inequality uses $\delta \leq 1/2$.

Now condition on the event $\bigcup_{i=1}^n E_i$: there exists an agent $i$ with $A_i = \varnothing$. For that agent, $v_i(G) = m$ and for any good $g \in G \setminus A_i$ we have $v_i(A_i \cup \{g\}) = 1$. But by the choice of $\alpha$ in (5),

$$\alpha \cdot \frac{v_i(G)}{n} = \frac{2n}{m} \cdot \frac{m}{n} = 2.$$

Hence $v_i(A_i \cup \{g\}) = 1 < 2 = \alpha \cdot \frac{v_i(G)}{n}$ for all $g \in G \setminus A_i$, so the allocation is not $\alpha$-PROP1. Therefore, the probability that RAND outputs an allocation that is not $\alpha$-PROP1 is at least

$$\Pr\left[\bigcup_{i=1}^n E_i\right] \geq \delta.$$

Finally, we show $\alpha = \Theta(1/\log(n/\delta))$. Since for $x \in (0,1)$, $x \leq -\log(1-x) \leq \frac{x}{1-x}$, plugging in $x = \frac{1}{n}$ gives us

$$\frac{1}{n} \leq \log\left(\frac{n}{n-1}\right) \leq \frac{1}{n-1}.$$

Thus, $m_0 = \frac{\log(n/(2\delta))}{\log(n/(n-1))}$ satisfies

$$(n-1)\log\left(\frac{n}{2\delta}\right) \leq m_0 \leq n\log\left(\frac{n}{2\delta}\right),$$

and therefore $m = \lfloor m_0 \rfloor = \Theta(n\log(n/\delta))$ (since $\delta \leq 1/2$ implies $\log(n/(2\delta)) = \Theta(\log(n/\delta))$). Consequently,

$$\alpha = \frac{2n}{m} = \Theta\left(\frac{1}{\log(n/\delta)}\right),$$

as claimed.

### C.3. Proof of Theorem 4.3

Fix an agent $i \in [n]$. If $v_i(G) = 0$, then $v_i(A_i) = 0 = (1-\varepsilon) \cdot v_i(G)/n$ always, and the claim holds trivially for this agent. Thus, assume $v_i(G) > 0$.

Under RAND, each good is allocated independently and uniformly at random to one of the $n$ agents. For each $j \in [m]$, let $X_j \sim \text{Bernoulli}(1/n)$ indicate that $g_j$ is allocated to agent $i$. Define $X_j' := 1 - X_j$, so $X_j' \sim \text{Bernoulli}((n-1)/n)$ indicates that $g_j$ is *not* allocated to $i$. Define

$$S_i := \sum_{j=1}^{m} v_i(g_j)X_j = v_i(A_i) \quad \text{and} \quad S_i' := \sum_{j=1}^{m} v_i(g_j)X_j'.$$

Since $X_j' = 1 - X_j$, we have that

$$S_i' = v_i(G) - S_i. \tag{10}$$

Moreover,

$$\mathbb{E}[S_i'] = \sum_{j=1}^{m} v_i(g_j)\mathbb{E}[X_j'] = \frac{n-1}{n}\sum_{j=1}^{m} v_i(g_j) = \frac{n-1}{n}v_i(G).$$

We will upper bound the probability that agent $i$ receives less than a $(1-\varepsilon)$ fraction of their proportional share:

$$\Pr\left[S_i < (1-\varepsilon) \cdot \frac{v_i(G)}{n}\right].$$

Using (10), this event is equivalent to

$$S_i' > v_i(G) - (1-\varepsilon) \cdot \frac{v_i(G)}{n} = \frac{n-1+\varepsilon}{n}v_i(G) = \mathbb{E}[S_i'] + \varepsilon \cdot \frac{v_i(G)}{n}.$$

Thus, letting $t := \varepsilon \cdot \frac{v_i(G)}{n}$,

$$\Pr\left[S_i < (1-\varepsilon) \cdot \frac{v_i(G)}{n}\right] = \Pr\left[S_i' > \mathbb{E}[S_i'] + t\right]. \tag{11}$$

Because the adversary is non-adaptive, the values $v_i(g_j)$ are fixed in advance (deterministic). RAND assigns each good independently, so the random variables $Y_j = v_i(g_j)X_j'$ are independent, allowing us to apply Bernstein's inequality. Thus, we now apply Bernstein's inequality (refer to Theorem C.2) to the sum $S_i' = \sum_{j=1}^{m} Y_j$ where $Y_j := v_i(g_j)X_j'$. First, each summand satisfies

$$Y_j - \mathbb{E}[Y_j] \leq Y_j \leq v_i(g_j) \leq \max_{g \in G} v_i(g),$$

so Bernstein applies with $b := \max_{g \in G} v_i(g)$. Second, since the $X_j'$ are independent Bernoulli$((n-1)/n)$,

$$\text{Var}(Y_j) = v_i(g_j)^2 \text{Var}(X_j') = v_i(g_j)^2 \cdot \frac{n-1}{n} \cdot \frac{1}{n} = v_i(g_j)^2 \cdot \frac{n-1}{n^2} \leq \frac{v_i(g_j)^2}{n}.$$

By independence,

$$\sigma^2 := \text{Var}(S_i') = \sum_{j=1}^{m} \text{Var}(Y_j) \leq \frac{1}{n} \sum_{j=1}^{m} v_i(g_j)^2 \leq \frac{\max_{g \in G} v_i(g)}{n} \sum_{j=1}^{m} v_i(g_j) = \frac{\max_{g \in G} v_i(g) \cdot v_i(G)}{n},$$

where we used $v_i(g_j)^2 \leq \max_{g \in G} v_i(g) \cdot v_i(g_j)$ (and $v_i(g_j) \geq 0$).

Applying Bernstein's inequality with this $b$ and $\sigma^2$ gives us

$$\Pr\left[S_i' > \mathbb{E}[S_i'] + t\right] \leq \exp\left(-\frac{t^2}{2\sigma^2 + 2bt/3}\right).$$

Substituting $t = \varepsilon \cdot \frac{v_i(G)}{n}$ and using $\sigma^2 \leq \frac{\max_{g \in G} v_i(g) \cdot v_i(G)}{n}$ and $b = \max_{g \in G} v_i(g)$,

$$2\sigma^2 + \frac{2}{3}bt \leq 2 \cdot \frac{\max_{g \in G} v_i(g) \cdot v_i(G)}{n} + \frac{2}{3} \max_{g \in G} v_i(g) \cdot \varepsilon \cdot \frac{v_i(G)}{n} \leq \frac{8}{3} \cdot \frac{\max_{g \in G} v_i(g) \cdot v_i(G)}{n},$$

where we used $\varepsilon \leq 1$ in the last inequality. Therefore,

$$\Pr\left[S_i' > \mathbb{E}[S_i'] + t\right] \leq \exp\left(-\frac{\varepsilon^2 \cdot \frac{v_i(G)^2}{n^2}}{\frac{8}{3} \cdot \frac{\max_{g \in G} v_i(g) \cdot v_i(G)}{n}}\right) = \exp\left(-\frac{3\varepsilon^2}{8} \cdot \frac{\frac{v_i(G)}{n}}{\max_{g \in G} v_i(g)}\right).$$

Since $\max_{g \in G} v_i(g) \leq \frac{3\varepsilon^2}{8\log(n/\delta)} \cdot \frac{v_i(G)}{n}$,

$$\frac{3\varepsilon^2}{8} \cdot \frac{\frac{v_i(G)}{n}}{\max_{g \in G} v_i(g)} \geq \log\left(\frac{n}{\delta}\right),$$

so

$$\Pr\left[S_i' > \mathbb{E}[S_i'] + t\right] \leq \exp\left(-\log\left(\frac{n}{\delta}\right)\right) = \frac{\delta}{n}.$$

Combining with (11), we get

$$\Pr\left[v_i(A_i) < (1-\varepsilon) \cdot \frac{v_i(G)}{n}\right] \leq \frac{\delta}{n}.$$

A union bound over all agents $i \in [n]$ implies that with probability at least $1 - \delta$,

$$v_i(A_i) \geq (1-\varepsilon) \cdot \frac{v_i(G)}{n} \quad \text{for all } i \in [n]. \tag{12}$$

Finally, (12) implies $(1-\varepsilon)$-PROP1: for any agent $i$, if $A_i = G$ we are done; otherwise pick any $g \in G \setminus A_i$, and

$$v_i(A_i \cup \{g\}) \geq v_i(A_i) \geq (1-\varepsilon) \cdot \frac{v_i(G)}{n}.$$

Thus the allocation is $(1-\varepsilon)$-PROP1 with probability at least $1 - \delta$.

## D. Omitted Proofs from Section 5

### D.1. Proof of Proposition 5.1

Suppose for a contradiction that there exists an online algorithm that achieves an approximation ratio of $\alpha$ for EF1, MMS, or PROPX. Let $v_i(g_1) = 1$ for all $i \in [n]$, and assume without loss of generality that agent 1 is allocated $g_1$.

We now define an instance with $m$ goods. Let $v_i(g_1) = 1$ for all $i \in [n]$, and for each $t \in \{2, \ldots, m\}$, the valuations are as follows:

$$v_1(g_t) = \begin{cases} 1 & \text{if } |A_1^t| = 1 \text{ or } t \leq n \\ 0 & \text{otherwise} \end{cases}$$

$$v_i(g_t) = \begin{cases} \varepsilon K^{t-2} & \text{if } |A_1^t| = 1 \text{ or } t \leq n \\ 0 & \text{otherwise} \end{cases}$$

where $m = \lceil \frac{n}{\alpha} \rceil + n + 2$, $K = \lceil \frac{3}{\alpha} \rceil$, and $\varepsilon = \frac{1}{K^{m-2}}$. Since every good has value at most 1 for all agents, this construction is consistent with the MIV prediction with $\mathbf{p} = (1, \ldots, 1)$.

We split our analysis into two cases.

**Case 1:** $|A_1| = 1$. Then, by the assumption that agent 1 is allocated $g_1$, agents' valuations for incoming goods must be as follows:

| $\mathbf{v}$ | $g_1$ | $g_2$ | $g_3$ | $\cdots$ | $g_m$ |
|---|---|---|---|---|---|
| 1 | ① | 1 | 1 | $\ldots$ | 1 |
| 2 | 1 | $\varepsilon$ | $\varepsilon K$ | $\ldots$ | $\varepsilon K^{m-2}$ |
| $\vdots$ | $\vdots$ | $\vdots$ | $\vdots$ | $\vdots$ | $\vdots$ |
| $n$ | 1 | $\varepsilon$ | $\varepsilon K$ | $\ldots$ | $\varepsilon K^{m-2}$ |

Note that since

$$m = \left\lceil \frac{n}{\alpha} \right\rceil + n + 2 \geq \frac{n}{\alpha} + n + 2, \tag{13}$$

we have that

$$\alpha \geq \frac{n}{m-n-2} > \frac{n}{m-n-1}. \tag{14}$$

Furthermore, as $|A_1| = 1$, we get that $v_1(A_1) = v_1(g_1) = 1$. We first show a contradiction to $\alpha$-EF1. Since $|A_1| = 1$, by the pigeonhole principle, there must exist an agent $i \neq 1$ that receives at least $\lceil \frac{m-1}{n-1} \rceil \geq \frac{m-1}{n-1} > \frac{m-1}{n}$ goods. Thus, there exists a good $g \in A_i$ such that

$$v_1(A_i \setminus \{g\}) \geq \frac{m-1}{n} - 1 = \frac{m-n-1}{n}.$$

Consequently, we get that

$$\frac{v_1(A_1)}{v_1(A_i \setminus \{g\})} \leq \frac{n}{m-n-1} < \alpha,$$

where the rightmost inequality follows from (14), giving us

$$v_1(A_1) < \alpha \cdot v_1(A_i \setminus \{g\}),$$

a contradiction.

Next, we show a contradiction to $\alpha$-MMS. Note that

$$\mathsf{MMS}_1 = \left\lfloor \frac{m}{n} \right\rfloor \geq \frac{m}{n} - 1 = \frac{m-n}{n}.$$

Then,

$$\frac{v_1(A_1)}{\mathsf{MMS}_1} \leq \frac{n}{m-n} < \alpha,$$

where again, the rightmost inequality follows from (14), giving us

$$v_1(A_1) < \alpha \cdot \mathsf{MMS}_1,$$

a contradiction.

Finally, we show a contradiction to $\alpha$-PROPX. From (13) and the fact that $\alpha \leq 1$, we get that $m \geq 2n + 2$.

Observe that for $g \in \arg\min_{g' \in G \setminus A_1} v_1(g')$, we have that

$$v_1(A_1 \cup \{g\}) = 1 + 1 = 2.$$

Together with the fact that $v_1(G) = m$, we obtain

$$
\begin{aligned}
\frac{n}{v_1(G)} \cdot v_1(A_1 \cup \{g\}) = \frac{2n}{m} &\leq \frac{2n}{2n+2} \\
&= \frac{n}{n+1} \\
&\leq \frac{n}{m-n-1} < \alpha,
\end{aligned}
$$

where the last inequality follows from (14). This gives us $v_i(A_1 \cup \{g\}) < \alpha \cdot \frac{v_1(G)}{n}$, a contradiction.

**Case 2:** $|A_1| > 0$. Let $g_t$ be the first good after $g_1$ that agent 1 receives. We further split our analysis into two sub-cases: (a) $t \leq n$ and (b) $t \geq n+1$.

**Case 2(a):** $t \leq n$. Then agents' valuations for incoming goods must be as follows:

| $\mathbf{v}$ | $g_1$ | $g_2$ | $\cdots$ | $g_t$ | $\cdots$ | $g_n$ |
|---|---|---|---|---|---|---|
| 1 | ① | 1 | $\cdots$ | ① | $\cdots$ | 1 |
| 2 | 1 | $\varepsilon$ | $\cdots$ | $\varepsilon K^{t-2}$ | $\cdots$ | $\varepsilon K^{n-2}$ |
| $\vdots$ | $\vdots$ | $\vdots$ | $\vdots$ | $\vdots$ | $\vdots$ | $\vdots$ |
| $n$ | 1 | $\varepsilon$ | $\cdots$ | $\varepsilon K^{t-2}$ | $\cdots$ | $\varepsilon K^{n-2}$ |

and if $n < m$, $v_i(g_j) = 0$ for all $i \in [n]$ and $j \in \{n+1, \ldots, m\}$. Since agent 1 receives two goods out of the first $n$ goods, by the pigeonhole principle, there necessarily exists an agent $i \in [n] \setminus \{1\}$ that does not receive any good out of the first $n$ goods, and $v_i(A_i) = 0$. Also, as $v_i(A_1 \setminus \{g\}) \geq \varepsilon K^{t-2}$ (for $g := \arg\max_{g \in A_1} v_i(g)$), we have that

$$
\frac{v_i(A_i)}{v_i(A_1 \setminus \{g\})} \leq \frac{0}{\varepsilon K^{t-2}} < \alpha,
$$

giving us $v_i(A_i) < \alpha \cdot v_i(A_1 \setminus \{g\})$, a contradiction to $\alpha$-EF1.

Furthermore, as $\mathsf{MMS}_i = v_i(g_2) = \varepsilon$, we have that

$$
\frac{v_i(A_i)}{\mathsf{MMS}_i} \leq \frac{0}{\varepsilon} < \alpha,
$$

giving us $v_i(A_i) < \alpha \cdot \mathsf{MMS}_i$, a contradiction to $\alpha$-MMS.

Next, we prove a contradiction to $\alpha$-PROPX. Again let $i$ be the agent that does not receive any good out of the first $n$ goods (as reasoned above). First observe that

$$
v_i(G) = 1 + \sum_{j=2}^{n} \varepsilon K^{j-2} = 1 + \frac{\varepsilon(K^{n-1} - 1)}{K - 1}.
$$

Also, since $K = \lceil \frac{3}{\alpha} \rceil$, it implies $\alpha > \frac{3}{K}$. Since $\alpha \leq 1$, it also means $K > 3$. Consequently, we have that

$$
3K^{n-3} \geq 3 \cdot 3^{n-3} = 3^{n-2} \geq n,
$$

giving us $n \leq 3K^{n-3}$, which is equivalent to

$$
\frac{n}{K^{n-2}} \leq \frac{3}{K} < \alpha.
$$

Moreover, for $g := \arg\min_{g' \in G \setminus A_i} v_i(g')$, we have $v_i(A_i \cup \{g\}) = 0 + v_i(g_2) = \varepsilon$.

Combining all the facts above, we get that

$$\frac{n \cdot v_i(A_i \cup \{g\})}{v_i(G)} = \frac{n \cdot \varepsilon}{1 + \frac{\varepsilon(K^{n-1}-1)}{K-1}}$$

$$\leq \frac{n \cdot \varepsilon}{\frac{\varepsilon(K^{n-1}-1)}{K-1}}$$

$$= \frac{n(K-1)}{K^{n-1}-1}$$

$$\leq \frac{n(K-1)}{K^{n-2}(K-1)}$$

$$= \frac{n}{K^{n-2}}$$

$$< \alpha,$$

giving us $v_i(A_i \cup \{g\}) < \alpha \cdot \frac{v_i(G)}{n}$, a contradiction.

**Case 2(b):** $t > n + 1$. Then agents' valuations for incoming goods must be as follows:

| $\mathbf{v}$ | $g_1$ | $g_2$ | $\cdots$ | $g_t$ |
|---|---|---|---|---|
| 1 | ①  | 1 | $\cdots$ | ①  |
| 2 | 1 | $\varepsilon$ | $\cdots$ | $\varepsilon K^{t-2}$ |
| $\vdots$ | $\vdots$ | $\vdots$ | $\vdots$ | $\vdots$ |
| $n$ | 1 | $\varepsilon$ | $\cdots$ | $\varepsilon K^{t-2}$ |

and if $t < m$, $v_i(g_j) = 0$ for all $i \in [n]$ and $j \in \{t+1, \ldots, m\}$.

Consider the set $\{g_{t+2-n}, \ldots, g_t\}$. Since the set contains $n-1$ goods and agent 1 is allocated $g_t$, by the pigeonhole principle, there exists an agent $i \in [n] \setminus \{1\}$ that does not receive any good in $\{g_{t+2-n}, \ldots, g_t\}$. Hence,

$$v_i(A_i) \leq \sum_{j=2}^{t+1-n} \varepsilon \cdot K^{j-2} = \sum_{j'=0}^{t-1-n} \varepsilon \cdot K^{j'} \leq \frac{\varepsilon(K^{t-n}-1)}{K-1} \leq \frac{2\varepsilon(K^{t-n}-1)}{K} \leq \frac{2}{K}\left(\varepsilon K^{t-n}\right).$$

Moreover, since $K = \lceil \frac{3}{\alpha} \rceil \geq \frac{3}{\alpha}$ and $\alpha \leq 1$, we know that $K \geq 2$, $\frac{1}{K-1} \leq \frac{2}{K}$, and $\alpha \geq \frac{3}{K} > \frac{2}{K}$.

We now show a contradiction to $\alpha$-EF1. Since $v_i(g_1) \geq v_i(g_t)$, we have that

$$v_i(A_1 \setminus \{g_1\}) = v_i(g_t) = \varepsilon K^{t-2} \geq \varepsilon K^{t-n},$$

where the last inequality follows from the fact that $n \geq 2$. This means there exists a $g \in A_1$ whereby

$$\frac{v_i(A_i)}{v_i(A_1 \setminus \{g\})} \leq \frac{\frac{2}{K}\left(\varepsilon K^{t-n}\right)}{\varepsilon K^{t-n}} = \frac{2}{K} < \alpha,$$

and implies $v_i(A_i) < \alpha \cdot v_i(A_i \setminus \{g\})$, a contradiction.

Furthermore, consider the set of $n$ goods $G' = \{g_{t+2-n}, \ldots, g_t\} \cup \{g_1\}$. We have that

$$\mathsf{MMS}_i \geq \min_{g \in G'} v_i(g) = v_i(g^{t_2-n}) = \varepsilon K^{t-n}.$$

Thus, we get

$$\frac{v_i(A_i)}{\mathsf{MMS}_i} \leq \frac{\frac{2}{K}\left(\varepsilon K^{t-n}\right)}{\varepsilon K^{t-n}} \leq \frac{2}{K} < \alpha,$$

giving us $v_i(A_i) < \alpha \cdot \mathsf{MMS}_i$, a contradiction.

Finally, for $g := \arg\min_{g' \in G \setminus A_i} v_i(g)$, we have that

$$v_i(A_i \cup \{g\}) \leq \frac{2}{K}\left(\varepsilon K^{t-n}\right) + \varepsilon K^{t-n} = \varepsilon K^{t-n}\left(\frac{2}{K} + 1\right)$$

and

$$v_i(G) = 1 + \frac{\varepsilon(K^{n-1}-1)}{K-1}$$

Then,

$$\frac{n \cdot v_i(A_i \cup \{g\})}{v_i(G)} = \frac{n\varepsilon K^{t-n}\left(\frac{2}{K}+1\right)}{1 + \frac{\varepsilon(K^{n-1}-1)}{K-1}} \leq \frac{n\varepsilon K^{t-n}\left(\frac{2}{K}+1\right)}{\frac{\varepsilon(K^{n-1}-1)}{K-1}}$$

$$= \frac{nK^{t-n}\left(\frac{2}{K}+1\right)(K-1)}{K^{n-1}-1}$$

$$\leq \frac{nK^{\lceil n/\alpha\rceil+2}\left(\frac{2}{K}+1\right)(K-1)}{K^{n-1}-1},$$

where the last inequality follows from the fact that $n+1 < t \leq m$ and hence

$$K^{t-n} \leq K^{m-n} \leq K^{\lceil n/\alpha\rceil+2}.$$

Now,

$$\frac{nK^{\lceil n/\alpha\rceil+2}\left(\frac{2}{K}+1\right)(K-1)}{K^{n-1}-1} < \frac{nK^{\lceil n/\alpha\rceil+3}}{K^{n-1}-1} \quad \left(\text{since } \left(\frac{2}{K}+1\right)(K-1) < K\right)$$

$$\leq 2nK^{\lceil n/\alpha\rceil+3-(n-1)} \quad \left(\text{since } K^{n-1}-1 \geq \frac{1}{2}K^{n-1}\right)$$

$$= 2nK^{\lceil n/\alpha\rceil+4-n}$$

$$\leq 2nK^{n/\alpha+5-n} \quad (\text{since } \lceil n/\alpha\rceil \leq n/\alpha+1)$$

$$= \frac{2n}{K^{n(1-1/\alpha)-5}}$$

$$< 2n\left(\frac{\alpha}{3}\right)^{n(1-1/\alpha)-5} \quad \left(\text{since } K > \frac{3}{\alpha}\right)$$

$$< \alpha.$$

This gives us $v_i(A_i \cup \{g\}) < \alpha \cdot v_i(G)$, which is a contradiction to $\alpha$-PROPX.

In both cases, we get a contradiction, and hence our result follows.

### D.2. Proof of Theorem 5.2

For each agent $i \in [n]$, note that $r_i$ is the earliest iteration (of the **while** loop in Line 3 of Algorithm 1) whereby agent $i$ values a good at 1, i.e., $v_i(g_{r_i}) = 1$ and for each $1 \leq t < r_i$, $v_i(g_t) < 1$.

Recall that for each $i \in [n]$ and $t \in \{1, \ldots, m\}$, $A_i^t$ denotes the bundle of agent $i$ after $g_t$ has been allocated. To simplify the analysis, we introduce new variables in place of those used in the algorithm (from Line 8 onwards). For each $t \in \{0, \ldots, m\}$, we denote

$$x_i^t = \begin{cases} \frac{1}{1+v_i(G^t)} & \text{if } t < r_i \\ \frac{1}{v_i(G^t)} & \text{otherwise} \end{cases}$$

and

$$y_i^t = \begin{cases} \frac{v_i(A_i^t)}{1+v_i(G^t)} & \text{if } t < r_i \\ \frac{v_i(A_i^t\setminus\{g_{r_i}\})}{v_i(G^t)} & \text{otherwise} \end{cases}$$

Note that $x_i^0 = 1$ and $y_i^0 = 0$ for all $i \in [n]$. Also, for each $i \in [n]$ and $t \in [m]$, we let

$$\phi_i^t := \frac{x_i^t}{(n^2+n+1)\cdot x_i^t + n^2 \cdot y_i^t - 1} \tag{15}$$

and

$$\Phi^t = \sum_{i\in[n]} \phi_i^t. \tag{16}$$

Then, it is easy to observe that Algorithm 1 is essentially an online algorithm that minimizes $\Phi^t$ at each round. We first present a lemma, which will be useful in our proof.

**Lemma D.1.** *For each $t \in \{0, \ldots, m\}$, if (i) $\phi_i^t \geq 0$ for all $i \in [n]$, and (ii) $\Phi^t \leq \Phi^0 = \frac{1}{n+1}$, then (1) for all $i \in [n]$, $x_i^t + y_i^t \geq \frac{1}{n^2}$, and (2) if $t = m$, the allocation $\mathcal{A}$ returned by Algorithm 1 satisfies $\frac{1}{n}$-PROP1.*

*Proof.* If $A_i = G$, we are done. Thus, assume $A_i \subsetneq G$. We first prove that for each agent $i \in [n]$, $x_i^t + y_i^t \geq \frac{1}{n^2}$. Consider any $t \in \{0, \ldots, m\}$. Suppose for a contradiction that

(i) $\phi_i^t \geq 0$ for all $i \in [n]$, and

(ii) $\Phi^t \leq \frac{1}{n+1}$,

but there exists some agent $j \in [n]$ such that $x_j^t + y_j^t < \frac{1}{n^2}$. Then, we have that

$$
\begin{aligned}
\phi_j^t &= \frac{x_j^t}{(n^2 + n + 1) \cdot x_j^t + n^2 \cdot y_j^t - 1} \quad \text{(by (15))} \\
&> \frac{x_j^t}{(n+1) \cdot x_j^t} \quad \text{(since } x_j^t + y_j^t < \frac{1}{n^2}) \\
&= \frac{1}{n+1} \\
&\geq \Phi^t \quad \text{(by (i))} \\
&= \sum_{i \in [n]} \phi_i^t \quad \text{(by 16)} \\
&\geq \phi_j^t, \quad \text{(since } \phi_i^t \geq 0 \text{ for all } i \in [n])
\end{aligned}
$$

thereby giving us a contradiction. Thus, for each agent $i \in [n]$, $x_i^t + y_i^t \geq \frac{1}{n^2}$.

Next, we show that if the two invariant holds, then when the algorithm terminates, the resulting allocation $\mathcal{A}$ satisfies $\frac{1}{n}$-PROP1.

Now, at the last iteration $t = m$, for each agent $i \in [n]$, it must be that $t \geq r_i$, and hence by definition we get

$$
x_i^m = \frac{1}{v_i(G^m)} \quad \text{and} \quad y_i^m = \frac{v_i(A_i^m \setminus \{g_{r_i}\})}{v_i(G^m)}.
$$

Since $G^m = G$ and $A_i^m = A_i$, we get

$$
x_i^m + y_i^m = \frac{1 + v_i(A_i \setminus \{g_{r_i}\})}{v_i(G)}. \tag{17}
$$

If $g_{r_i} \in A_i$, then there exists a good $g \in G \setminus A_i$ (note this must be true because we assumed $A_i \subsetneq G$) such that $v_i(A_i \cup \{g\}) \geq v_i(A_i) = v_i(A_i \setminus \{g_{r_i}\}) + 1$; and if $g_{r_i} \notin A_i$, then $v_i(A_i \cup \{g_{r_i}\}) = v_i(A_i) + 1 = v_i(A_i \setminus \{g_{r_i}\}) + 1$. In both cases, we have that there exists a $g \in G \setminus A_i$ such that

$$
v_i(A_i \cup \{g\}) = v_i(A_i \setminus \{g_{r_i}\}) + 1.
$$

Consequently,

$$
\begin{aligned}
v_i(A_i \cup \{g\}) &\geq v_i(A_i \setminus \{g_{r_i}\}) + 1 \\
&= (x_i^m + y_i^m) \cdot v_i(G) \quad \text{(by (17))} \\
&\geq \frac{1}{n^2} \cdot v_i(G),
\end{aligned}
$$

where the last inequality follows from the fact we proved earlier in this lemma. Observe that the expression is equivalent to

$$
v_i(A_i \cup \{g\}) \geq \frac{1}{n} \cdot \frac{v_i(G)}{n}
$$

for some $g \in G \setminus A_i$, and $\frac{1}{n}$-PROP1 is satisfied. $\qquad \square$

Next, we prove that the algorithm has two invariants.

**Lemma D.2.** *For each $t \in \{0, \ldots, m\}$, the following hold:*

   *(i) $\phi_i^t \geq 0$ for all $i \in [n]$, and*

   *(ii) $\Phi^t \leq \frac{1}{n+1}$.*

*Proof.* We prove the lemma by induction.

First, consider the base case when $t = 0$. For every agent $i \in [n]$, $G^0 = \varnothing$ and thus, $v_i(G^0) = v_i(A_i^0) = v_i(A_i) = 0$. This gives us $x_i^0 = \frac{1}{1+0} = 1$ and $y_i^0 = \frac{0}{1+0} = 0$. Consequently, we get that

$$\phi_i^0 = \frac{1}{(n^2 + n + 1) - 1} = \frac{1}{n^2 + n} \geq 0,$$

satisfying (i). Moreover, we have that

$$\Phi^0 = \sum_{i \in [n]} \phi_i^0 = \sum_{i \in [n]} \frac{1}{n^2 + n} = \frac{n}{n^2 + n} = \frac{1}{n+1},$$

satisfying (ii). Thus, the base case holds.

Next, we prove the inductive step. Suppose that for every $k \in \{0, \ldots, T-1\}$, we have that

$$\phi_i^k \geq 0 \text{ for all } i \in [n] \quad \text{and} \quad \Phi^k \leq \frac{1}{n+1}. \tag{18}$$

We want to prove that

$$\phi_i^{k+1} \geq 0 \text{ for all } i \in [n] \quad \text{and} \quad \Phi^{k+1} \leq \frac{1}{n+1}.$$

For each agent $i \in [n]$, let

- $y_i^+, \phi_i^+$ be the values of $x_i^{k+1}, y_i^{k+1}, \phi_i^{k+1}$ respectively agent $i$ is allocated good $g_{k+1}$ (i.e., $g_{k+1} \in A_i^{k+1}$), and
- $y_i^-, \phi_i^-$ be the values of $x_i^{k+1}, y_i^{k+1}, \phi_i^{k+1}$ respectively if agent $i$ is not allocated good $g_{k+1}$ (i.e., $g_{k+1} \notin A_i^{k+1}$).

We first prove that for each agent $i \in [n]$,

$$\phi_i^{k+1} \geq 0 \quad \text{and} \quad \phi_i^+ + (n-1) \cdot \phi_i^- - n \cdot \phi_i^k \leq 0. \tag{19}$$

We split our analysis into two cases, depending on whether $k + 1 = r_i$ or $k + 1 \neq r_i$.

**Case 1:** $k + 1 = r_i$. This means that $v_i(g_{k+1}) = 1$. Moreover, since $k < r_i$, we get that

$$x_i^k = \frac{1}{1 + v_i(G^k)} \quad \text{and} \quad y_i^k = \frac{v_i(A_i^k)}{1 + v_i(G^k)}.$$

Thus,

$$\begin{aligned} x_i^{k+1} &= \frac{1}{v_i(G^{k+1})} = \frac{1}{v_i(g_{k+1}) + v_i(G^k)} \\ &= \frac{1}{1 + v_i(G^k)} \quad \text{(since } v_i(g_{k+1}) = 1) \\ &= x_i^k. \end{aligned}$$

Similarly, we have that

$$y_i^+ = y_i^- = \frac{v_i(A_i^{k+1} \setminus \{g_{k+1}\})}{v_i(G^{k+1})} = \frac{v_i(A_i^k)}{1 + v_i(G^k)} = y_i^k.$$

Also, we have that $x_i^{k+1} = x_i^k$ and $y_i^{k+1} = y_i^k$, giving us

$$\phi_i^+ = \phi_i^- = \phi_i^{k+1} - 1.$$

Consequently, we get

$$\phi_i^+ + (n-1) \cdot \phi_i^- - n \cdot \phi_i^k = 0,$$

which proves the second part of (19). Moreover, since $\phi_i^k \geq 0$ (from (18)), we have that $\phi_i^+ = \phi_i^- \geq 0$, which means $\phi_i^{k+1} \geq 1 > 0$, giving us the first part of (19).

**Case 2:** $k + 1 \neq r_i$. To simplify our argument, we define an auxiliary valuation function $v_i'$ (which behaves more like a variable rather than a valuation function) defined as follows for $k$ and $k + 1$.

$$v_i'(G^k) = \begin{cases} 1 + v_i(G^k) & \text{if } k+1 < r_i; \\ v_i(G^k) & \text{otherwise.} \end{cases}$$

$$v_i'(G^{k+1}) = \begin{cases} 1 + v_i(G^{k+1}) & \text{if } k+1 < r_i; \\ v_i(G^{k+1}) & \text{otherwise.} \end{cases}$$

We also note that $v_i(A_i^k \setminus \{g_{r_i}\}) = v_i(A_i^k)$ if $k + 1 < r_i$. This gives us

$$x_i^k = \frac{1}{v_i'(G^k)} \quad \text{and} \quad y_i^k = \frac{v_i(A_i^k \setminus \{g_{r_i}\})}{v_i'(G^k)}.$$

Now, denote $\varepsilon := \frac{v_i(g_{k+1})}{v_i'(G^k)}$. Then, since $v_i(g_{k+1}) \leq p_i = v_i^{\max} = 1$, we have that

$$\varepsilon = \frac{v_i(g_{k+1})}{v_i'(G^k)} \leq \frac{1}{v_i'(G^k)} = x_i^k. \tag{20}$$

Thus, we have

$$
\begin{aligned}
x_i^{k+1} &= \frac{1}{v_i'(G^{k+1})} \\
&= \frac{1}{v_i(g_{k+1}) + v_i'(G^k)} \\
&= \frac{1}{\varepsilon \cdot v_i'(G^k) + v_i'(G^k)} \quad \text{(by definition of } \varepsilon) \\
&= \frac{1}{(1+\varepsilon) \cdot v_i'(G^k)} \\
&= \frac{x_i^k}{1+\varepsilon} \quad \text{(by definition of } x_i^k).
\end{aligned}
$$

Also, get that

$$
\begin{aligned}
y_i^+ &= \frac{v_i(A_i^k \setminus \{g_{r_i}\}) + v_i(g_{k+1})}{v_i'(G^{k+1})} \\
&= \frac{v_i(A_i^k \setminus \{g_{r_i}\}) + v_i(g_{k+1})}{v_i'(G^k) + v_i(g_{k+1})} \\
&= \frac{v_i(A_i^k \setminus \{g_{r_i}\}) + \varepsilon \cdot v_i'(G^k)}{v_i'(G^k) + \varepsilon \cdot v_i'(G^k)} \quad \text{(by definition of } \varepsilon) \\
&= \frac{v_i(A_i^k \setminus \{g_{r_i}\}) + \varepsilon \cdot v_i'(G^k)}{(1+\varepsilon) \cdot v_i'(G^k)} \\
&= \frac{y_i^k + \varepsilon}{1+\varepsilon} \quad \text{(by definition of } y_i^k)
\end{aligned}
$$

and

$$
\begin{aligned}
y_i^- &= \frac{v_i(A_i^k \setminus \{g_{r_i}\})}{v_i'(G^{k+1})} \\
&= \frac{v_i(A_i^k \setminus \{g_{r_i}\})}{v_i'(G^k) + v_i(g_{k+1})} \\
&= \frac{v_i(A_i^k \setminus \{g_{r_i}\})}{v_i'(G^k) + \varepsilon \cdot v_i'(G^k)} \quad \text{(by definition of } \varepsilon) \\
&= \frac{v_i(A_i^k \setminus \{g_{r_i}\})}{(1 + \varepsilon) \cdot v_i'(G^k)} \\
&= \frac{y_i^k}{1 + \varepsilon} \quad \text{(by definition of } y_i^k).
\end{aligned}
$$

We also have

$$
\begin{aligned}
\phi_i^+ &= \frac{x_i^{k+1}}{(n^2 + n + 1) \cdot x_i^{k+1} + n^2 \cdot y_i^+ - 1} \\
&= \frac{\frac{x_i^k}{1+\varepsilon}}{(n^2 + n + 1) \cdot \frac{x_i^k}{1+\varepsilon} + n^2 \cdot \frac{y_i^k + \varepsilon}{1+\varepsilon} - 1} \\
&= \frac{x_i^k}{(n^2 + n + 1) \cdot x_i^k + n^2 \cdot y_i^k + \varepsilon \cdot (n^2 - 1) - 1}
\end{aligned}
$$

and

$$
\begin{aligned}
\phi_i^- &= \frac{x_i^{k+1}}{(n^2 + n + 1) \cdot x_i^{k+1} + n^2 \cdot y_i^- - 1} \\
&= \frac{\frac{x_i^k}{1+\varepsilon}}{(n^2 + n + 1) \cdot \frac{x_i^k}{1+\varepsilon} + n^2 \cdot \frac{y_i^k}{1+\varepsilon} - 1} \\
&= \frac{x_i^k}{(n^2 + n + 1) \cdot x_i^k + n^2 \cdot y_i^k - \varepsilon - 1}.
\end{aligned}
$$

Now, denote $\Delta := (n^2 + n + 1) \cdot x_i^k + n^2 \cdot y_i^k - 1$. For any $i \in [n]$, we can then express $\phi_i^+$, $\phi_i^-$, and $\phi_i^k$ as the following:

$$
\phi_i^+ = \frac{x_i^k}{\Delta + \varepsilon \cdot (n^2 - 1)}, \quad \phi_i^- = \frac{x_i^k}{\Delta - \varepsilon}, \quad \phi_i^k = \frac{x_i^k}{\Delta}.
$$

Moreover, by (18), we know that $\phi_i^k \geq 0$ for all $i \in [n]$ and $\Phi^k \leq \frac{1}{n+1}$. Applying Lemma D.1, we get that

$$
n^2 \cdot (x_i^k + y_i^k) \geq 1.
$$

This means

$$
\begin{aligned}
\Delta &= (n^2 + n + 1) \cdot x_i^k + n^2 \cdot y_i^k - 1 \\
&= n^2 \cdot (x_i^k + y_i^k) + (n + 1) \cdot x_i^k - 1 \\
&\geq 1 + (n + 1) \cdot x_i^k - 1 \\
&= (n + 1) \cdot x_i^k. \tag{21}
\end{aligned}
$$

Then, we know that

- $\Delta > 0$ since $x_i^k > 0$;

- $\Delta - \varepsilon > 0$ since $\varepsilon \leq x_i^k$ (by (20)); and
- $\Delta + \varepsilon \cdot (n^2 - 1) > 0$ since $\varepsilon \geq 0$.

This gives us $\phi_i^+ \geq 0$ and $\phi_i^- \geq 0$. Then, by the definition of $\phi_i^+$ and $\phi_i^-$, we get that $\phi_i^{k+1} \geq 0$, thereby proving the first part of (19).

Now, by (21) and (20), we have that

$$\Delta \geq (n+1) \cdot x_i^k \geq (n+1) \cdot \varepsilon,$$

which gives us

$$\Delta \geq (n+1) \cdot \varepsilon = \frac{n(n^2 - 1)}{n(n-1)} \cdot \varepsilon.$$

Equivalently, we get

$$\Delta \cdot (n^2 - n) \geq n(n^2 - 1) \cdot \varepsilon.$$

Then, multiplying $\varepsilon$ on both sides, we obtain

$$
\begin{aligned}
0 &\geq n(n^2 - 1) \cdot \varepsilon^2 - \Delta \cdot \varepsilon \cdot (n^2 - n) \\
&= n(n^2 - 1) \cdot \varepsilon^2 - \Delta \cdot \varepsilon \cdot (n^2 - 1) + \Delta \cdot \varepsilon \cdot (n - 1) \\
&= \Delta \cdot (\Delta - \varepsilon) + (n \cdot \varepsilon - \Delta)(\Delta + \varepsilon \cdot (n^2 - 1)) \\
&\geq \frac{\Delta \cdot (\Delta - \varepsilon) + (n \cdot \varepsilon - \Delta)(\Delta + \varepsilon \cdot (n^2 - 1))}{(\Delta + \varepsilon \cdot (n^2 - 1)) \cdot \Delta \cdot (\Delta - \varepsilon)} \\
&= \frac{1}{\Delta + \varepsilon \cdot (n^2 - 1)} + \frac{n \cdot \varepsilon - \Delta}{\Delta \cdot (\Delta - \varepsilon)} \\
&= \frac{1}{\Delta + \varepsilon \cdot (n^2 - 1)} + \frac{n - 1}{\Delta - \varepsilon} - \frac{n}{\Delta}.
\end{aligned}
\tag{22}
$$

Finally,

$$
\begin{aligned}
\phi_i^+ + (n-1) \cdot \phi_i^- - n \cdot \phi_i^k &= \frac{x_i^k}{\Delta + \varepsilon \cdot (n^2 - 1)} + \frac{(n-1) \cdot x_i^k}{\Delta - \varepsilon} - \frac{n \cdot x_i^k}{\Delta} \\
&= x_i^k \left( \frac{1}{\Delta + \varepsilon \cdot (n^2 - 1)} + \frac{n-1}{\Delta - \varepsilon} - \frac{n}{\Delta} \right) \\
&\leq 0,
\end{aligned}
$$

where the inequality follows from (22) and the fact that $x_i^k \geq 0$, thereby proving the second part of (19).

It remains to prove that $\Phi^{k+1} \leq \frac{1}{n+1}$ to complete the inductive step. Now, we have that

$$
\begin{aligned}
0 &\geq \sum_{i \in [n]} \left( \phi_i^+ + (n-1) \cdot \phi_i^- - n \cdot \phi_i^k \right) \quad \text{(by (19))} \\
&= \sum_{i \in [n]} \phi_i^+ + (n-1) \cdot \sum_{i \in [n]} \phi_i^- - n \cdot \sum_{i \in [n]} \phi_i^k \\
&= \sum_{i \in [n]} \phi_i^+ + (n-1) \cdot \sum_{i \in [n]} \phi_i^- - n \cdot \Phi^k \\
&= \sum_{i \in [n]} \left( -\Phi^k + \phi_i^+ + \sum_{j \in [n] \setminus \{i\}} \phi_j^- \right).
\end{aligned}
$$

This necessarily means that there exists an agent $i^* \in [n]$ such that $-\Phi^k + \phi_{i^*}^+ + \sum_{j \in [n] \setminus \{i^*\}} \phi_j^- \leq 0$. If $g_{k+1}$ is allocated to agent $i^*$, then for all $i \in [n]$, we have that

$$
\phi_i^{k+1} = \begin{cases} \phi_{i^*}^+ & \text{if } i = i^*; \\ \phi_i^- & \text{otherwise.} \end{cases}
\tag{23}
$$

Let $\Phi_{i^*}^{k+1}$ be the value of $\Phi^{k+1}$ if $g_{k+1}$ was allocated to agent $i^*$. This means that

$$\Phi_{i^*}^{k+1} = \phi_{i^*}^+ + \sum_{i \in [n] \setminus \{i^*\}} \phi_i^-.$$

Since $\Phi_{i^*}^{k+1} - \Phi^k \leq 0$ and $\Phi^k \leq \frac{1}{n+1}$ (by (18)), we get that

$$\Phi_{i^*}^{k+1} \leq \Phi^k \leq \frac{1}{n+1}.$$

Finally, since the algorithm allocates $g_{k+1}$ to the agent that minimizes $\Phi^{k+1}$ (Line 18 of Algorithm 1), we know that $\Phi^{k+1} \leq \Phi_{i^*}^{k+1} \leq \frac{1}{n+1}$, as desired.

We have shown both the base case and inductive step, and thus, by induction, the lemma holds. $\qquad \square$

Thus, the invariants hold, and our result follows.

### D.3. Proof of Theorem 5.3

Without loss of generality, we can assume that $p_i = 1$ for each $i \in [n]$. Also let $r_i$ be the earliest timestep whereby agent $i$ values a good at least $1 - \varepsilon$, i.e., $v_i(g_{r_i}) \geq 1 - \varepsilon$ and for each $1 \leq t < r_i$, $v_i(g_t) < 1 - \varepsilon$.

Now, for each agent $i \in [n]$, define $\delta_i := 1 - v_i(g_{r_i})$ (so that $0 \leq \delta_i \leq \varepsilon$) and create the augmented valuation function $v_i'$ defined as follows:

$$v_i'(g_t) = \begin{cases} 1 & \text{if } t = r_i \\ v_i(g_t) & \text{otherwise.} \end{cases}$$

This also means

$$v_i'(g_{r_i}) = 1 = v_i(g_{r_i}) + \delta_i. \tag{24}$$

Run the $\alpha$-PROP1 algorithm on this sequence of goods, but using the augmented valuation functions $v_1', \ldots, v_n'$ instead. This returns an $\alpha$-PROP1 allocation with respect to the augmented valuation functions. Let $\mathcal{A} = (A_1, \ldots, A_n)$ be the returned allocation. By definition, for each agent $i \in [n]$, there exists some good $g \in G \setminus A_i$ such that

$$v_i'(A_i \cup \{g\}) \geq \alpha \cdot \frac{v_i'(G)}{n}. \tag{25}$$

For each agent $i \in [n]$, we have that $v_i'(g_t) = v_i(g_t)$ for all $t \in [m] \setminus \{r_i\}$ and $v_i'(g_{r_i}) = v_i(g_{r_i}) + \delta_i$ (by (24)). This gives us

$$v_i'(G) = v_i(G) + \delta_i \tag{26}$$

and for any $g \in G \setminus A_i$,

$$v_i'(A_i \cup \{g\}) \leq v_i(A_i \cup \{g\}) + \delta_i, \tag{27}$$

with the above being an equality if $g_{r_i} \notin A_i \cup \{g\}$. Together with the fact that $\delta_i \leq \varepsilon < 1$, this also implies there exists a $g \in G \setminus A_i$ such that

$$\frac{v_i(A_i \cup \{g\})}{1 - \delta_i} \geq \frac{v_i'(A_i \cup \{g\}) - \delta_i}{1 - \delta_i} \geq 1. \tag{28}$$

The rightmost inequality follows from the fact that $v_i'(A_i \cup \{g\}) \geq 1$, which is true if $g_{r_i} \in A_i$, otherwise we can choose $g$ to be $g_{r_i}$.

Also, since $\delta \leq \varepsilon$, we get that $\frac{1}{1-\delta_i} \leq \frac{1}{1-\varepsilon}$ and

$$\frac{\delta_i}{1 - \delta_i} \leq \frac{\varepsilon}{1 - \varepsilon}. \tag{29}$$

Now, for each $i \in [n]$, there exists a $g \in G \setminus A_i$ such that

$$
\begin{aligned}
\alpha \cdot \frac{v_i(G)}{n} &= \frac{\alpha}{n} \cdot (v'_i(G) - \delta_i) \quad \text{(by (26))} \\
&\leq v'_i(A_i \cup \{g\}) - \frac{\alpha}{n} \cdot \delta_i \quad \text{(by (25))} \\
&\leq (v_i(A_i \cup \{g\}) + \delta_i) - \frac{\alpha}{n} \cdot \delta_i \quad \text{(by (27))} \\
&= v_i(A_i \cup \{g\}) + \left(1 - \frac{\alpha}{n}\right) \cdot \delta_i \\
&\leq v_i(A_i \cup \{g\}) + \left(1 - \frac{\alpha}{n}\right) \cdot \frac{v_i(A_i \cup \{g\}) \cdot \delta_i}{1 - \delta_i} \quad \text{(by (28))} \\
&= \left(1 + \left(1 - \frac{\alpha}{n}\right) \cdot \frac{\delta_i}{1 - \delta_i}\right) \cdot v_i(A_i \cup \{g\}) \\
&\leq \left(1 + \left(1 - \frac{\alpha}{n}\right) \cdot \frac{\varepsilon}{1 - \varepsilon}\right) \cdot v_i(A_i \cup \{g\}) \quad \text{(by (29))} \\
&= \frac{1 - \frac{\alpha\varepsilon}{n}}{1 - \varepsilon} \cdot v_i(A_i \cup \{g\}) = \frac{\alpha}{\beta} \cdot v_i(A_i \cup \{g\}).
\end{aligned}
$$

This gives us

$$
v_i(A_i \cup \{g\}) \geq \beta \cdot \frac{v_i(G)}{n},
$$

as desired.

