# OpenReview forum: "Approximate Proportionality in Online Fair Division"
_ICML.cc/2026/Conference — ICML 2026 regular_

### Official Review · Reviewer_9zPR · 2026-03-11

**Soundness:** 4
**Presentation:** 3
**Significance:** 2
**Originality:** 3
**Overall Recommendation:** 4
**Confidence:** 3

**Summary:**

The authors study fair division under the following specifications: (1) items arrive online (i.e., over time), (2) the employed fairness notion is PROP1, and (3) this notion of fairness is relaxed by only aiming for a multiplicative approximation of PROP1.
They present three greedy algorithms and argue that for each of them, there exists a sequence of arriving goods under which the algorithm fails to produce an $\alpha$-PROP1 allocation for any $\alpha$.
Based on the exhibited shortcomings of greedy strategies, the authors restrict the problem setting further: (4) They consider non-adaptive adversaries (i.e., the algorithm's past decisions are not accessible to the adversary, so he cannot change his strategy dynamically), and (5) they assume maximum item value (MIV) predictions (i.e., the algorithm is given a prediction of the value that each agent assigns to the good that he prefers most).

The authors show that under a non-adaptive adversary, allocating items uniformly at random yields $\alpha$-PROP1 with high probability. Assuming MIV predictions, they strengthen this result to $1/n$-PROP1.

**Compliance With Llm Reviewing Policy:**

Affirmed.

**Final Justification:**

I stay with my original recommendation of weak acceptance, as I rate the significance and novelty as good but not great. The paper itself is written nicely and the authors promised to address the minor comments I had.

**Key Questions For Authors:**

I would appreciate some more background on MIV predictions.
1. How well-studied are they in the online EF1-setting?
2. Are there any realistic szenarios in practice where MIV predictions arise naturally?
3. Would it be useful to consider minimum-item-values or average-item-values instead?

Moreover, some background on greedy algorithms in online approximate settings would be useful.

4. Are the strategies 1 to 3 motivated by prior work?

**Limitations:**

yes

**Strengths And Weaknesses:**

Analysing fair division under the restrictions (1) to (3) is well-motivated, since exact online PROP1 has been studied before but is known to not necessarily exist, which is a reasonable starting point for an approximative analysis. The authors then showcase the failure of three greedy algorithms. In Section 1.1, it is claimed that the strategies that they implement correspond to "common heuristics". While each strategy seems reasonable to me, it would have been nice to elaborate further in what sense these strategies are common. Has there been prior work that employed such strategies? If so, have they been successful? It would have been interesting to sketch at the beginning of Section 3 which kind of greedy strategies lie behind the positive approximate results for MMS and EF1.

Subsequently, the authors use the shortcoming of the aforementioned greedy algorithms to set the stage for restrictions (4) and (5), but to me, the greedy algorithms only indicate that some restriction may be helpful, whereas choosing precisely (4) and (5) feels arbitrary. As for restriction (4), the study of non-adversary settings is at least somewhat motivated in the introduction by the non-existence of approximations for other fairness notions such as MMS and EF1. However, for restriction (5), there is no reference to prior work where MIV predictions have been employed in the online EF setting. In line 337, the authors claim that maximum value information is more realistic than normalization or total-value information, which is a claim that should be backed-up with arguments.

The proofs are quite detailed which helped a lot in verifying their correctness. I have not spotted any soundness issues, but did not check all proofs in detail. In line 360, I would have appreciated some more intuition on why the unary prediction vector can be assumed w.l.o.g.
It is hard to assess the originality of the employed proof techniques, since the authors do not comment on prior techniques in the domain. In my understanding, the paper does not propose any new techniques, but adapting the existing ones seems challenging enough.

The paper is overall well-written, with the exception of the beginning of Section 5, where several sentences are grammatically incorrect.
Some minor comments:
- Section 1.2: $n$ should be defined before it appears in the approximative factor. Also, there is a typo at the end of the paragraph.
- In definition 2.3, $X_j$ should be $A_j$.
- line 225: Speaking of an "unallocated good" here does not match the formal definition, which accounts for all goods that are already allocated but just not to agent $i$.
- The potential function in line 366 includes terms that are only properly defined in the appendix (for instance $r_i$). They should be at least explained intuitively.

---

> ### Author Rebuttal · Authors · 2026-03-29
>
> Thank you for your time and effort in the reviewing process, and for the positive assessment! Please find our response below.
>
> ---
> > I would appreciate some more background on MIV predictions. How well-studied are they in the online EF1-setting? Are there any realistic scenarios in practice where MIV predictions arise naturally? Would it be useful to consider minimum-item-values or average-item-values instead?
>
> MIV predictions are not meant to be a standard assumption from prior online EF1 work; rather, they are the deliberately **lightweight** advice model we chose here. To the best of our knowledge, the closest prior information models in online fair division are quite different: [1] assume normalized valuations/known **total** value and study online MMS; [2] study normalization information and frequency predictions; [3] consider a fully informed temporal model with complete knowledge of future items and valuations; and the recent online EFX with predictions work of [4] assumes a full predicted valuation vector for future goods. Given these, our use of MIV is intentional: it asks for only one coarse scalar per agent (an upper bound on the value of a single future item) rather than total values, value multisets, or per-item forecasts.
>
> Practically, MIV is as a per-agent peak-value or scale estimate. Online fair division and allocation settings already arise in applications such as ad allocation, recommendation exposure, moderation workload, and compute-slot assignment, where decisions are sequential and irrevocable. In such settings, it is natural to model the case where a platform can estimate a conservative upper envelope on an agent’s best plausible per-item value from historical data, even when it cannot predict the entire future value vector or the exact total future value. That is the modeling role MIV is meant to capture.
>
> We also do not think minimum item value or average item value predictions play the same role for PROP1. PROP1 is repaired by adding one missing good, so the structurally relevant statistic is an upper bound on the value of one good. A minimum item value is often vacuous (e.g., a single zero-valued/dummy good makes it 0) while an average item value can be made arbitrarily small by many zero-valued dummy arrivals and therefore does not control the one-good repair term. These statistics may be interesting for other models, but they are not aligned with the structure of PROP1 or with the potential/slack accounting used in our analysis.
>
> ---
> >  In Section 1.1, it is claimed that the strategies that they implement correspond to "common heuristics". While each strategy seems reasonable to me, it would have been nice to elaborate further in what sense these strategies are common. Has there been prior work that employed such strategies? If so, have they been successful? It would have been interesting to sketch at the beginning of Section 3 which kind of greedy strategies lie behind the positive approximate results for MMS and EF1.
>
> The first strategy is commonly used for utilitarian welfare-maximizing guarantees; the second strategy is common in offline fair division (e.g., to achieve EFX when valuations are identical in the offline setting [5], in the online setting to achieve EF1 for any number of agents when valuations are identical [2,3]; in the fully-informed online fair division setting to get temporal EF1 under generalized binary valuations [3]; and in many other fair division settings); and the third strategy is the most direct greedy attempt to optimize the PROP1 objective itself. We will clarify this.
>
> ---
> > In line 360, I would have appreciated some more intuition on why the unary prediction vector can be assumed w.l.o.g. It is hard to assess the originality of the employed proof techniques, since the authors do not comment on prior techniques in the domain. In my understanding, the paper does not propose any new techniques, but adapting the existing ones seems challenging enough.
>
> This is because $\alpha$-PROP1 is homogeneous in each agent's valuation scale. Dividing all of agent $i$’s valuations by $p_i$ scales both sides of the $\alpha$-PROP1 inequality by the same factor and simultaneously normalizes the predicted maximum to $1$, so the guarantee, feasibility and approximation ratio are unchanged. We will mention this explicitly.
>
> ---
> We also appreciate the minor comments and will correct the noted typos in the revision.
>
> ---
> [1] S. Zhou, R. Bai, X. Wu. Multi-agent online scheduling: MMS allocations for indivisible items. ICML, 2023.
>
> [2] T.Y. Neoh, J. Peters, and N. Teh. Online fair division with additional information. arXiv, 2025
>
> [3] E. Elkind, A. Lam, M. Latifian, T.Y. Neoh, N. Teh. Temporal fair division of indivisible items. AAMAS, 2025
>
> [4] T. Melissourgos, N. Protopapas. Online EFX Allocations with Predictions. arXiv, 2025
>
> [5] B. Plaut, T. Roughgarden. Almost envy-freeness with general valuations. SIAM Journal on Discrete Mathematics, 2020

---

> > ### Author Rebuttal · Reviewer_9zPR · 2026-04-02
> >
> > I thank the author(s) for their detailed replies to all questions

---

> > > ### Author Response · Authors · 2026-04-07
> > >
> > > We sincerely appreciate your follow-up and are glad that our clarifications have adequately addressed your concerns! Thank you again for your positive recommendation and thoughtful feedback, and we hope you can recommend our work for acceptance.

---

### Official Review · Reviewer_2fJf · 2026-03-12

**Soundness:** 2
**Presentation:** 2
**Significance:** 1
**Originality:** 2
**Overall Recommendation:** 2
**Confidence:** 4

**Summary:**

This paper studies the online fair allocation with online indivisible goods. This paper proposes three simple greedy algorithms and shows that these three algorithms can not produce $\alpha$-PROP1 allocations for any $\alpha > 0$. This paper studies the algorithm that allocate online items uniformly at random and show this random allocation can guarantee some PROP1 approximation allocation with a high probability. Moreover, if the algorithm has the access to the maximum good value in the future, this paper introduces an algorithm that returns a $1/n$-PROP1 allocation while no algorithm can return $\alpha$-EF1, $\alpha$-MMS or $\alpha$-PROPX for any $\alpha >0$.

**Compliance With Llm Reviewing Policy:**

Affirmed.

**Key Questions For Authors:**

1.	Can the authors explain more about the significance of the first result?
2.	Why does this paper not provide a deterministic algorithm based on the analysis of randomly allocating goods? I believe there is such an deterministic algorithm based on the random algorithm.

**Limitations:**

Section 3 introduces three greedy algorithms and presents a hardness result, but the role of these results in the overall framework is somewhat unclear. It would be helpful to clarify their practical or conceptual significance.

**Strengths And Weaknesses:**

Strength:
1.	This paper provides some negative results for online allocation even if the algorithm know the maximum good value in the future, which yields the hardness of the online fair allocation with online goods.
2.	With knowing the maximum good value, this paper presents a non-trivial algorithm that returns a $1/n$-PROP1 allocation.

Weakness:
1.	I think the first result for three greedy algorithms is confused and does not surprise me. When I read the abstract of this paper, the description about the first result really confused me. I would say the contribution of this paper will not decrease if deleting the result for three greedy algorithms.
2.	The technique for the second result of randomized allocation is common, which uses the measure concretrationto give some high probability guarantee. Moreover, since randomly allocate the online goods can guarantee a PROP1 approximation allocation with a very high probability, why there is no result for deterministic algorithms that can guarantee the same approximate ratio? I believe there is such an deterministic algorithm since just by randomly. allocating the goods, I can get a “good” PROP1 approximate ratio.
3.	 I think the third result is a little bit weak since the fairness notion is approximate PROP1 instead of approximate PROP, and the result is only $1/n$ which does not surprise me.
4.	All the techniques in these paper are common techniques. I do not feel that I learned a lot after reading this paper.

---

> ### Author Rebuttal · Authors · 2026-03-29
>
> Thank you for your time and effort in the reviewing process. Please find our response below.
>
> ---
>
> > Sec. 3 introduces three greedy algorithms and presents a hardness result, but the role of these results in the overall framework is somewhat unclear. It would be helpful to clarify their practical or conceptual significance. Can the authors explain more about the significance of the first result?
>
> As motivated in Sec. 1.1 (and again at the end of Sec. 3), Sec. 3 plays an structural role in our paper: it shows that purely local greedy decisions fail completely against adaptive adversaries, even for the weakest proportionality benchmark. This is important because these strategies correspond to the most natural and widely used heuristics (e.g., welfare maximization, balancing allocations, or directly optimizing the PROP1 objective). The result therefore explains why one must move beyond greedy approaches --- either to randomness (Sec. 4) or predictions (Sec. 5). We will clarify this motivation more explicitly in the introduction and at the start of Sec. 3.
>
> ---
>
> > Why does this paper not provide a deterministic algorithm based on the analysis of randomly allocating goods? I believe there is such an deterministic algorithm based on the random algorithm.
>
> We believe that Theorem 4.1 does **not** imply the existence of a deterministic algorithm with a comparable PROP1 approximation. The high-probability guarantee is taken over the algorithm's internal randomness, which an oblivious adversary does not observe. Such guarantees hold **for each fixed input sequence**, but the successful random realization may depend on that sequence. For instance, the randomness required to succeed on sequence $A$ may differ from the randomness required to succeed on sequence $B$; there is no guarantee that the same random seed works for both sequences $A$ and $B$. In contrast, a deterministic algorithm would require a single rule that succeeds uniformly over all sequences. Establishing such a result would therefore be strictly stronger than Theorem 4.1 and would resolve the open question highlighted in our conclusion.
>
> More broadly, it is well known that deterministic guarantees do not always match randomized ones even against oblivious adversaries. For example, in the classical paging problem, the optimal deterministic competitive ratio is $k$, while randomized algorithms achieve $\Theta(\log k)$. This gap arises because randomness mitigates adversarial effects that exploit predictable behavior. Our setting exhibits a similar structure: outcomes depend critically on the arrival order, and deterministic policies can be systematically skewed, whereas randomization prevents such persistent imbalance.
>
> ---
>
> > I think the third result is a little bit weak since the fairness notion is approximate PROP1 instead of approximate PROP, and the result is only $1/n$ which does not surprise me.
>
> We respectfully disagree that the result is weak because it targets approximate PROP1 rather than approximate PROP.
>
> Firstly, even in offline settings, PROP is too strong for indivisible goods: even with two agents and one positively valued good, one agent necessarily receives value $0$, so no positive multiplicative PROP guarantee is possible and approximate PROP is not a meaningful target. This is why the literature studies relaxations for indivisible goods, and our introduction already explains why PROP1 is the right proportional benchmark here.
>
> Secondly, we showed (Theorem 5.1) that even with perfect predictions, notions such as EF1, MMS, and PROPX (which are stronger than PROP1 and weaker than PROP) are inapproximable. This also naturally implies PROP is also inapproximable, since PROP implies PROPX. Thus, PROP1 is the natural frontier.
>
> Likewise, the significance of the $1/n$ factor is not that it is the final answer, but that it is the **first finite worst-case approximation known for any PROP-type notion under adaptive adversaries with such weak predictive information**. Tightening the dependence on $n$ is exactly the main open problem left by our paper.
>
> ---
>
> > All the techniques in these paper are common techniques. I do not feel that I learned a lot after reading this paper.
>
> We agree that several ingredients (e.g., concentration bounds, potential-based analysis) are classical **in isolation**. Our contribution is not a new standalone technique, but rather a complete characterization of the approximability landscape for online PROP1, including:
> - a barrier result for natural greedy methods
> - near-tight guarantees for random allocation
> - a learning-augmented algorithm with robustness guarantees
>
> As noted by reviewer 9zPR, "the paper does not propose any new techniques, but adapting the existing ones seems challenging enough".

---

### Official Review · Reviewer_mvV5 · 2026-03-14

**Soundness:** 4
**Presentation:** 4
**Significance:** 2
**Originality:** 3
**Overall Recommendation:** 5
**Confidence:** 3

**Summary:**

This paper studies the approximability of Proportionality up to one good (PROP1) in the challenging online fair division setting, where indivisible goods arrive sequentially and must be allocated immediately and irrevocably. The paper first demonstrates a strong negative result: three standard greedy allocation rules fail to provide any multiplicative PROP1 approximation against an adaptive adversary.

To overcome this theoretical barrier, the paper explores two practical relaxations: (i) Randomization against Non-adaptive Adversaries. It proves that a simple uniformly random allocation (RAND) achieves a high-probability PROP1 approximation against oblivious inputs. Further, when individual items are ``small'' relative to an agent's fair share, RAND achieves near-PROP1 fairness; (ii) Learning-Augmented Algorithms: By incorporating Maximum Item Value (MIV) predictions, the paper designs an algorithm that guarantees a $1/n$-PROP1 allocation even against adaptive adversaries. Crucially, they show this guarantee degrades under one-sided prediction errors. In contrast, other fairness notions (EF1, MMS, and PROPX) remain strictly inapproximable even with perfect MIV predictions.

**Compliance With Llm Reviewing Policy:**

Affirmed.

**Final Justification:**

Authors responded to my questions well and I maintain my positive score.

**Key Questions For Authors:**

Please refer to the detailed questions provided in the Strengths and Weaknesses above. Regarding my suggestion to investigate theoretical lower bounds: I acknowledge that deriving a strict lower bound for algorithms using MIV predictions might be highly challenging and potentially out of scope for a short rebuttal. If a formal mathematical proof is not feasible, could the authors instead discuss the fundamental hardness of this problem? Specifically, what are the main issues that make establishing a tight lower bound difficult in this setting?

**Limitations:**

Yes.

**Strengths And Weaknesses:**

- Soundness: The theoretical results of the paper appear to be reliable and technically sound. The core claims are supported by mathematical proofs. The analytical techniques are appropriate, covering the adversarial constructions for the greedy baselines, the probability bounds for the randomized allocation, and the potential function analysis for the learning-augmented algorithm. The authors also clearly state their assumptions and honestly address the limitations of their work, explicitly noting that the optimal dependency on the number of agents remains an open problem.
    - Presentation: The paper is well-written, logically organized, and easy to follow. The narrative arc is highly effective, building from the failure of natural greedy baselines to the constructive positive results using randomization and MIV predictions.
    - Significance: The theoretical significance of this paper is clear, as it successfully obtains the boundary between impossibility and achievability for PROP1 in online fair division. However, the practical significance of the proposed methods seems somewhat limited. My questions are listed below:

        1)  Investigate Lower Bounds: The 1/n-PROP1 guarantee provided by Algorithm 1 is practically very weak, especially in real-world systems where the number of agents n is large. The authors acknowledge that the optimal dependence on n remains an open problem. Establishing a theoretical lower bound for algorithms using MIV predictions would clarify whether 1/n is a fundamental mathematical limit of the problem.
        2)  Evaluate the Fairness-Efficiency Tradeoff: Does overemphasizing worst-case fairness lead to severely degraded overall social welfare? I suggest the authors to add some numerical experiments to empirically evaluate this tradeoff. The authors could plot a Pareto frontier comparing Algorithm 1 against the three greedy baselines. For instance, setting the X-axis as total social welfare (Efficiency) and the Y-axis as the PROP1 ratio (Fairness). This visualization would provide a much more intuitive understanding of the algorithm's practical costs and benefits.
        3)  Discuss Two-Sided Errors: The robustness guarantee relies strictly on a one-sided error assumption, meaning predictions can only overestimate. In real-world forecasting, two-sided errors are inevitable (e.g., an unforeseen high-value item arrives, causing an underestimation). Briefly discussing how Algorithm 1 behaves under two-sided errors would significantly enhance the paper's relevance to real-world deployment.

    - Originality: The originality of this work is solid. Applying the learning-augmented algorithmic framework to bypass existing impossibility results in online fair division is a valid and effective angle. However, while this specific application to PROP1 is novel, the general approach of using single-value predictions to overcome worst-case bounds is somewhat a standard technique in the online algorithms literature. Therefore, the conceptual novelty, while technically sound, is somewhat incremental and does not introduce a fundamentally new theoretical paradigm.

---

> ### Author Rebuttal · Authors · 2026-03-29
>
> Thank you for your time and effort in the reviewing process, and for the very positive assessment!  Please find our response below.
>
> ---
>
> > Does overemphasizing worst-case fairness lead to severely degraded overall social welfare? I suggest the authors to add some numerical experiments to empirically evaluate this tradeoff.
>
> We thank the reviewer for the suggestion and agree that evaluating the fairness-efficiency tradeoff would provide additional practical insight. Our current focus in this work is on worst-case guarantees, where greedy baselines can be driven to arbitrarily poor PROP1 under adaptive adversaries (Sec. 3), so the primary question we are investigating here is what can be guaranteed at all.
>
> Nevertheless, we ran the requested fairness-efficiency comparison between Alg. 1 and the three greedy baselines. Following the MIV-normalized view used by Alg. 1, we normalize each agent’s values so that their maximum item value is 1, measure fairness by the realized PROP1 ratio, and measure efficiency by utilitarian welfare normalized by the ex-post offline optimum. Empirically, Alg. 1 lies on or very near the Pareto frontier. In dense and correlated cases, it achieved exact PROP1 on all sampled instances we tested, while its welfare was only modestly below Greedy-1 (for example, at $n=8, m=40$: uniform instances 1.000 PROP1 / 0.891 welfare for Alg. 1 vs 0.734 / 0.945 for Greedy-1; dense instances 1.000 / 0.936 vs 0.447 / 0.968; correlated instances 1.000 / 0.916 vs 0.722 / 0.974). At the same time, Alg. 1 consistently outperformed Greedy-2 and Greedy-3 in these cases, achieving the same or better fairness with substantially higher welfare. On easy specialist instances, Greedy-1 can have higher welfare with essentially no fairness loss, suggesting that the welfare price of worst-case robustness is mainly relevant on competitive instances. We also stress-tested on the explicit counterexample families underlying our negative results; there the targeted greedy baseline’s PROP1 ratio collapses sharply, whereas Alg. 1 stays well above its worst-case guarantee and is often welfare-optimal or near-optimal. We will include a brief analysis of this in our revision.
>
> ---
>
> > Discuss Two-Sided Errors: The robustness guarantee relies strictly on a one-sided error assumption, meaning predictions can only overestimate. Briefly discussing how Alg. 1 behaves under two-sided errors would significantly enhance the paper's relevance to real-world deployment.
>
> Bounded two-sided multiplicative errors are already covered by our framework after a simple calibration step. Suppose the raw predictor satisfies $p_i \in [(1-\rho)v_i^{\max},(1+\rho) v_i^{\max}]$ for each agent $i$. If we conservatively inflate the prediction to $\tilde p_i := \frac{p_i}{1-\rho}$, then $\tilde p_i$ never underestimates the true MIV, and moreover $v_i^{\max}\in [\frac{1-\rho}{1+\rho}\tilde p_i,\ \tilde p_i]$.
> Thus the inflated predictions satisfy exactly our one-sided error model with effective error $\varepsilon=\frac{2\rho}{1+\rho}$.
>
> Therefore Thm. 5.3 (and hence the Alg. 1 robustness guarantee in Cor. 5.4) applies directly after this upward calibration. In this sense, our one-sided model should be viewed as the calibrated form of bounded two-sided noise, rather than as excluding it. We stated the result in one-sided form because that is the way the prediction is used in the analysis: as a conservative upper envelope for the single repair item in PROP1. What is genuinely harder is uncalibrated underestimation with no known error envelope, since a single unexpected high-value item can invalidate the slack accounting used by the proof.
> We will clarify this.
>
> ---
>
> > I acknowledge that deriving a strict lower bound for algorithms using MIV predictions might be highly challenging and potentially out of scope for a short rebuttal. If a formal mathematical proof is not feasible, could the authors instead discuss the fundamental hardness of this problem? Specifically, what are the main issues that make establishing a tight lower bound difficult in this setting?
>
> We agree that understanding lower bounds is an important direction. While a tight bound appears technically challenging, we can provide intuition for why this problem is fundamentally difficult.
>
> Standard lower bound constructions in online fair division rely on hiding whether a large future item exists. However, MIV predictions reveal precisely the scale of any future item value, which breaks standard hardness constructions for PROP1. Any tight lower bound would therefore need to fix the entire MIV vector while encoding hardness in the residual tail and arrival order, which is substantially more subtle.
>
> Our Prop. 5.1 already provides a strong partial lower bound that even with perfect MIV predictions, stronger notions (EF1, MMS, PROPX) remain inapproximable, suggesting that our positive result is highly specific to PROP1.
>
> We will add a paragraph clarifying this hardness intuition.

---

> > ### Author Rebuttal · Reviewer_mvV5 · 2026-04-01
> >
> > Thanks to the authors for their detailed response; I maintain my positive score.

---

> > > ### Author Response · Authors · 2026-04-07
> > >
> > > We sincerely appreciate your follow-up and are glad that our clarifications have adequately addressed your concerns! Thank you again for your positive recommendation and thoughtful feedback, and we hope you can recommend our work for acceptance.

---

### Official Review · Reviewer_V9ge · 2026-03-16

**Soundness:** 3
**Presentation:** 3
**Significance:** 3
**Originality:** 3
**Overall Recommendation:** 4
**Confidence:** 4

**Summary:**

The paper studies approximate guarantees for PROP1 in online fair division with indivisible goods.
Main results are
* Negative results for natural greedy strategies against adaptive adversaries.
* High probability approximate PROP1 guarantees for uniform random allocation against non-adaptive adversaries.
* Almost PROP1 result in the small items regime
* An online algorithm in the learning-augmented framework that achieves $1/n$-PROP1 under adaptive adversaries

**Compliance With Llm Reviewing Policy:**

Affirmed.

**Key Questions For Authors:**

No specific question

**Limitations:**

Discussed above

**Strengths And Weaknesses:**

## Strength
The obtained results are clean and comprehensive. The paper identifies a natural gap in the literature, namely PROP1 rather than EF1/MMS, and resolves it under several settings. The learning-augmented result is also well-motivated and fits ICML well.

## Weakness

I don't find specific weakness in the paper. I would say the the high-probability guarantee for RAND and the $1/n$ guarantee under learning-augmented setup are mathematically nontrivial, but perhaps practically less interesting.
I'm also not entirely sure if the paper is the best fit for ICML, though it is justified by the additional learning-augmented result.

---

> ### Author Rebuttal · Authors · 2026-03-27
>
> Thank you for your time and effort in the reviewing process, and for the positive assessment!
>
> We would like to clarify the practical relevance and positioning within the ML literature, and will revise our paper accordingly to emphasize these connections.
>
> From an ML perspective, our setting directly models sequential allocation under uncertainty with predictions, which arises in applications such as ad allocation, recommendation systems, and online marketplaces (as discussed in our introduction). In such systems, decisions must be made irrevocably while relying on coarse forecasts, making fairness guarantees under uncertainty a central concern.
>
> Our contributions are aligned with recent work in learning-augmented algorithms, which has become an active area in ICML/NeurIPS. In particular, prior works in this space [1,2,3] study how predictions can improve allocation or optimization objectives, but do not address individual-level fairness guarantees under adversarial arrivals. Our results show that even extremely weak predictions (MIV) qualitatively change what is achievable for fairness. Beyond the learning-augmented perspective, fair division papers have also routinely been published in venues such as ICML/NeurIPS, e.g., see [4,5,6,7].
>
> ---
>
> [1] Fabian Spaeh and Alina Ene. Online Ad Allocation with Predictions. NeurIPS 2023.
>
> [2] Ilan Reuven Cohen, Alon Eden, Talya Eden, and Arsen Vasilyan. Plant-and-Steal: Truthful Fair Allocations via Predictions. NeurIPS 2024.
>
> [3] Ilan Reuven Cohen and Debmalya Panigrahi. A Learning-Augmented Approach to Online Allocation Problems. NeurIPS, 2025.
>
> [4] Gerdus Benadè, Daniel Halpern, and Alexandros Psomas. Dynamic Fair Division with Partial Information. NeurIPS, 2022.
>
> [5] Shengwei Zhou, Rufan Bai, and Xiaowei Wu. Multi-agent Online Scheduling: MMS Allocations for Indivisible Items. ICML, 2023
>
> [6] Bo Li, Fangxiao Wang, and Shiji Xing. Settling the Maximin Share Fairness for Scheduling among Groups of Machines. ICML, 2025.
>
> [7] Ariel D. Procaccia, Benjamin Schiffer, and Shirley Zhang. Honor Among Bandits: No-Regret Learning for Online Fair Division. NeurIPS, 2024.

---

> > ### Author Rebuttal · Reviewer_V9ge · 2026-04-04
> >
> > My questions have been resolved and will maintain my score at the moment.

---

> > > ### Author Response · Authors · 2026-04-07
> > >
> > > We sincerely appreciate your follow-up and are glad that our rebuttal has fully resolved all your concerns! Thank you again for your positive recommendation and thoughtful feedback, and we hope you can recommend our work for acceptance.

---

### Decision · Program_Chairs · 2026-04-30

**Decision:**

Accept (regular)

**Comment:**

The paper studies the approximability of PROP1 in online fair division, establishing impossibility results for greedy baselines, high-probability guarantees for random allocation, and a learning-augmented algorithm achieving 1/n-PROP1 with MIV predictions. Three of the four reviewers supported acceptance, finding the results technically sound and the paper well-written.

Reviewer 2fJf raised concerns about significance, but did not acknowledge the rebuttal or engage in discussion despite multiple requests, so I have discounted this review. Several of their criticisms also reflected misunderstandings of the setting (e.g., suggesting that a randomized guarantee straightforwardly implies a deterministic one).